# Structural snapshots of Xer recombination reveal activation by synaptic complex remodeling and DNA bending

Aleksandra Bebel[1], Ezgi Karaca[1], Banushree Kumar[1], W Marshall Stark[2], Orsolya Barabas[1]*

[1]Structural and Computational Biology Unit, European Molecular Biology Laboratory, Heidelberg, Germany; [2]Institute of Molecular, Cell and Systems Biology, University of Glasgow, Glasgow, United Kingdom

**Abstract** Bacterial Xer site-specific recombinases play an essential genome maintenance role by unlinking chromosome multimers, but their mechanism of action has remained structurally uncharacterized. Here, we present two high-resolution structures of *Helicobacter pylori* XerH with its recombination site DNA *dif*$_H$, representing pre-cleavage and post-cleavage synaptic intermediates in the recombination pathway. The structures reveal that activation of DNA strand cleavage and rejoining involves large conformational changes and DNA bending, suggesting how interaction with the cell division protein FtsK may license recombination at the septum. Together with biochemical and in vivo analysis, our structures also reveal how a small sequence asymmetry in *dif*$_H$ defines protein conformation in the synaptic complex and orchestrates the order of DNA strand exchanges. Our results provide insights into the catalytic mechanism of Xer recombination and a model for regulation of recombination activity during cell division.

*For correspondence: barabas@embl.de

**Competing interests:** The authors declare that no competing interests exist.

## Introduction

In organisms with circular genomes, homologous recombination-mediated repair behind a stalled replication fork can join the two nascent daughter chromosomes, resulting in a chromosome dimer (*Barre et al., 2001*). Dimer formation prohibits proper segregation of the genetic information at cell division (*Figure 1A*), and must be repaired to produce viable progeny. In bacteria and archaea, chromosome dimers are monomerized by members of a large family of tyrosine recombinases, the Xer recombinases (*Blakely et al., 1991*, *1993*). These enzymes act by promoting recombination between two DNA sites, called *dif*. The *dif* site is normally present in a single copy at the replication terminus (*Carnoy and Roten, 2009*; *Kuempel et al., 1991*), but it is duplicated in chromosome dimers, so that intramolecular recombination results in separation ('resolution') of the two chromosome copies (*Figure 1A*). Removal of the *xer* genes or the *dif* site results in increased DNA content, activation of the SOS response, cell filamentation, and cell death (*Britton and Grossman, 1999*; *Debowski et al., 2012a*; *Hendricks et al., 2000*; *Pérals et al., 2000*; *Val et al., 2008*). Besides chromosome dimer resolution, Xer recombinases can support plasmid resolution and mobilization of the cholera toxin phage CTXϕ and pathogenicity islands (*Das et al., 2013*; *Fischer et al., 2010*).

In *E. coli*, Xer recombination is carried out by cooperation of two similar enzymes XerC and XerD (37% identity) that act together to bind and recombine *dif* sites (*Blakely et al., 1993*). Many other organisms (including *Lactococcus*, *Helicobacter*, *Campylobacter* spp. and archaea) employ a single Xer recombinase system (*Carnoy and Roten, 2009*; *Cortez et al., 2010*; *Debowski et al., 2012a*; *Le Bourgeois et al., 2007*; *Leroux et al., 2013*), with a prime example, XerH/*dif*$_H$, found in *Helicobacter pylori*, a gastric pathogen implicated in peptic ulcer disease and gastric cancer.

**eLife digest** Similar to humans, bacteria store their genetic material in the form of DNA and arrange it into structures called chromosomes. In fact, most bacteria have a single circular chromosome. Bacteria multiply by simply dividing in two, and before that happens they must replicate their DNA so that each of the newly formed cells receives one copy of the chromosome.

Occasionally, mistakes during the DNA replication process can cause the two chromosomes to become tangled with each other; this prevents them from separating into the newly formed cells. For instance, the chromosomes can become physically connected like links in a chain, or merge into one long string. This kind of tangling can result in cell death, so bacteria encode enzymes called Xer recombinases that can untangle chromosomes. These enzymes separate the chromosomes by cutting and rejoining the DNA strands in a process known as Xer recombination.

Although Xer recombinases have been studied in quite some detail, many questions remain unanswered about how they work. How do Xer recombinases interact with DNA? How do they ensure they only work on tangled chromosomes? And how does a protein called FtsK ensure that Xer recombination takes place at the correct time and place?

Bebel et al. have now studied the Xer recombinase from a bacterium called *Helicobacter pylori*, which causes stomach ulcers, using a technique called X-ray crystallography. This enabled the three-dimensional structure of the Xer recombinase to be visualized as it interacted with DNA to form a Xer-DNA complex. Structures of the enzyme before and after it cut the DNA show that Xer-DNA complexes first assemble in an inactive state and are then activated by large conformational changes that make the DNA bend. Bebel et al. propose that the FtsK protein might trigger these changes and help to bend the DNA as it activates Xer recombination.

Further work showed that the structures could be used to model and understand Xer recombinases from other species of bacteria. The next step is to analyze how FtsK activates Xer recombinases and to see if this process is universal amongst bacteria. Understanding how this process can be interrupted could help to develop new drugs that can kill harmful bacteria.

Xer recombinases are members of the tyrosine site-specific recombinase superfamily, a large group of enzymes that catalyze DNA breakage and rejoining using a conserved tyrosine nucleophile (*Grindley et al., 2006*; *Guo et al., 1997*; *Midonet and Barre, 2014*; *Nunes-Düby et al., 1998*). Tyrosine recombinases promote various programmed DNA rearrangements including the monomerization of phage, plasmid and chromosome multimers, resolution of hairpin telomeres, and the movement of virulence and antibiotic resistance carrying integrative mobile genetic elements (including phages and transposons)(*Grindley et al., 2006*; *Jayaram et al., 2015*; *Midonet and Barre, 2014*). In addition, tyrosine recombinases (as exemplified by Cre and Flp) provide powerful genetic engineering tools that are widely used to carry out mutagenesis and DNA insertion in eukaryotic chromosomes (*Nagy, 2000*; *Turan et al., 2011*).

Tyrosine recombinases share a common chemical mechanism that involves step-wise breakage and exchange of four DNA strands in pairs, proceeding through a characteristic four-way Holliday junction (HJ) DNA intermediate (*Figure 1B*)(*Gopaul and Duyne, 1999*; *Grindley et al., 2006*; *Holliday, 2007*). They cut each DNA strand with a polarity creating a covalent 3' phosphotyrosyl protein-DNA linkage and a free 5' hydroxyl group. The DNA ends then go on to join with the complementary ends of the partner DNA strand generating the recombined products. All DNA cleavage and rejoining reactions take place in an ordered protein-DNA synaptic complex, comprising four recombinase molecules holding the two recombination partner DNA molecules together.

An unusual feature of Xer recombination at *dif* is that it requires an accessory factor, FtsK (*Aussel et al., 2002*; *Debowski et al., 2012a*; *Le Bourgeois et al., 2007*; *Leroux et al., 2013*; *Nolivos et al., 2010*; *Steiner et al., 1999*). This DNA motor protein localizes to the bacterial cell division septum and contributes to segregating the sister chromosomes into the daughter cells by translocating towards their replication termini. On chromosome dimers, FtsK stops at the Xer-bound *dif* sites and activates recombination, triggering resolution of the dimers to monomers (*Aussel et al., 2002*; *Grainge et al., 2011*; *May et al., 2015*). Without FtsK, Xer-*dif* synaptic

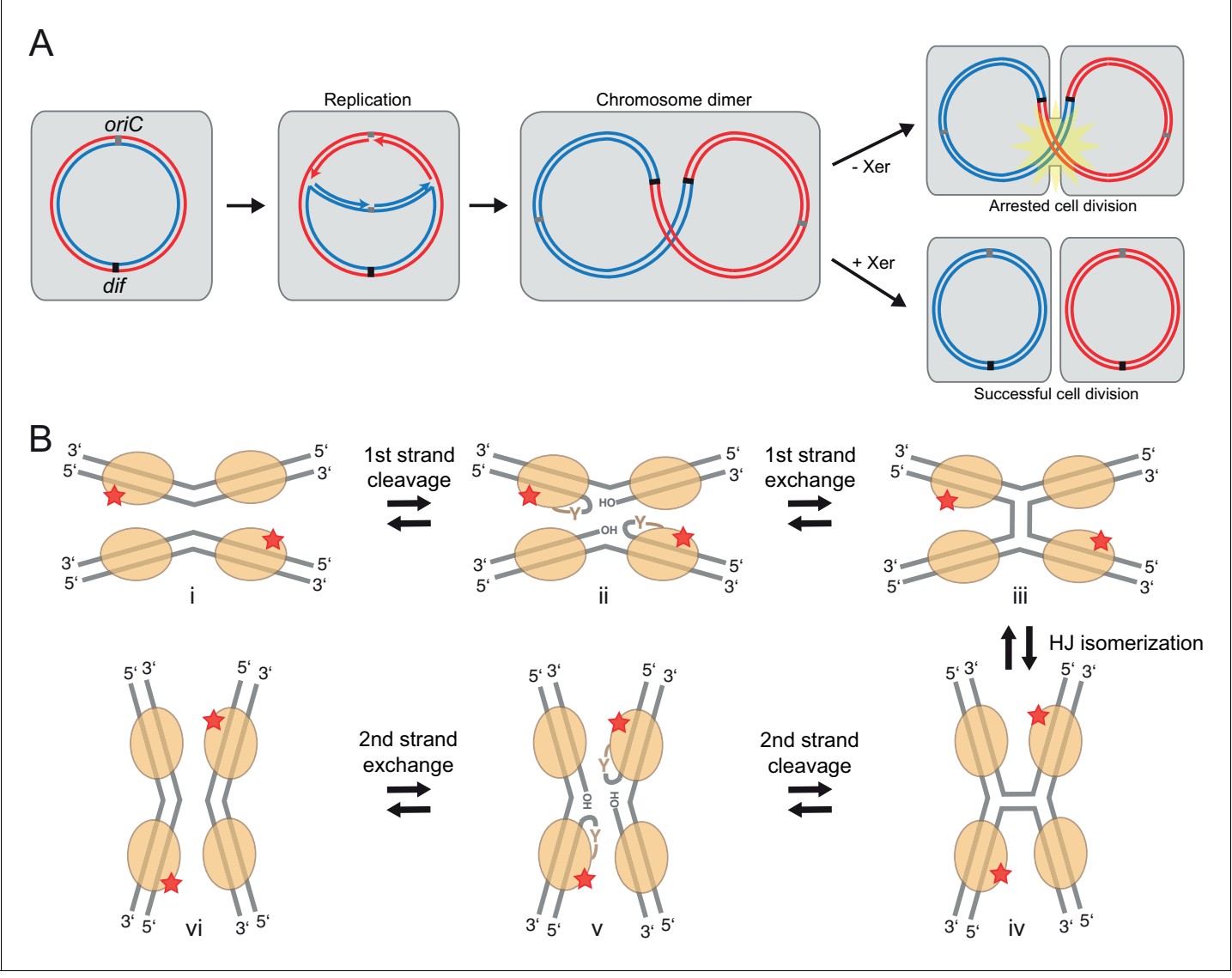

**Figure 1.** Xer recombination. (**A**) The role of Xer recombination in the maintenance of bacterial chromosomes. Homologous recombination behind a stalled replication fork can result in a chromosome dimer. Xer recombinases monomerize these to rescue healthy genome segregation. The absence of Xer leads to cell division arrest and cell death. (**B**) Schematic representation of tyrosine site-specific recombination. Two recombinase monomers (beige ovals) bind one specific DNA site (grey) and two such sites are aligned in antiparallel in a tetrameric synaptic complex (**i**). The catalytic tyrosines of two symmetry-related protomers (red star) cleave one strand of each DNA, creating a covalent 3' phosphotyrosyl bond and a free 5' hydroxyl group (**ii**). The latter then attacks the phosphotyrosyl bond of the partner DNA, forming the HJ intermediate (**iii**). Following an isomerization step, the second pair of protomers becomes catalytically active (**iv**), leading to a reciprocal set of cleavage and strand exchange reactions that resolve the HJ and generate the recombined DNA products (**v–vi**). Only two protein subunits are active in the tetramer at a time ('half-of-the-sites reactivity'), strictly ordering the chemical steps to ensure faithful progression of the recombination reaction to the desired products. Note that the DNA substrates are drawn with the strand going 3' to 5' on the top.

complexes are formed, but do not lead to final recombination products (*Aussel et al., 2002*; *Diagne et al., 2014*; *Grainge et al., 2011*; *Zawadzki et al., 2013*). Regulation by FtsK is critical to ensure that Xer recombination takes place only in the correct spatial and temporal context – at the division septum, when genome replication has been completed – thereby ensuring faithful genome segregation.

Here, we present two crystal structures of the *H. pylori* XerH recombinase in complex with its recombination site *dif$_H$*. Together with associated biochemical and in vivo data, these first Xer-DNA

complex structures shed light on potential regulatory mechanisms of the recombination pathway. Remarkably, the overall shape and DNA conformation of initially formed XerH-$dif_H$ synaptic complexes is considerably different from those of other tyrosine recombinases, such as Cre-$loxP$ that has served as a model system for the family. The unanticipated conformation of the pre-cleavage synaptic complex suggests a possible model for why Xer proteins require external activation, and in comparison with the post-cleavage complex structure provides clues for how FtsK might stimulate recombination activity. Our structures provide a resource to construct models for other Xer synaptic complexes, including that of the heterotetrameric *E. coli* XerC/D system.

## Results

### Structure of the XerH-$dif_H$ synaptic complex

The current mechanistic model of Xer recombination conforms to the tyrosine recombinase paradigm, which is supported by extensive biochemical and structural studies on Xer and other systems (reviewed in: [*Grindley et al., 2006*; *Midonet and Barre, 2014*; *Van Duyne, 2001*]). This model proposes a step-wise process that starts with two Xer monomers binding to each *dif* site, which then interact to form a synaptic complex (*Figure 1B, i*) (*Blakely et al., 1993*). Here, two Xer protomers each cleave one strand of one *dif* site (*Figure 1B ii*) (*Blakely et al., 1997*; *Gopaul and Duyne, 1999*; *Guo et al., 1997*), and the broken strands are exchanged and rejoined creating the HJ intermediate (*Figure 1B, iii*) (*Gopaul et al., 1998*). Then the second Xer pair performs cleavage and strand exchange on the other strand pair, completing recombination (*Figure 1B, v–vi*). Based on available synaptic complex structures of other tyrosine recombinases, it was hypothesized that the *dif* DNA is bent upon synapsis, creating a square planar DNA arrangement that is then maintained throughout the recombination reaction to allow energetically inexpensive exchange of DNA strands (*Gopaul et al., 1998*; *Guo et al., 1997*, *1999*). However, in the absence of direct structural data the exact architecture of Xer-DNA complexes has remained unknown. Previous DNA-free crystal structures of Xer recombinases (*Jo et al., 2016*; *Serre et al., 2013*; *Subramanya et al., 1997*) showed a domain arrangement that is incompatible with DNA binding. Another puzzling aspect of the mechanism concerned activation by FtsK. In early work, it was noted that in the absence of *E. coli* FtsK, HJ formation was catalysed by XerC, leading to the hypothesis that XerC/D-*dif* synaptic complexes assemble preferentially with XerC in an active conformation (*Barre et al., 2000*). Subsequently, FtsK was shown to interact directly with the C-terminal domain of XerD (*Keller et al., 2016*; *Yates et al., 2003*, *2006*), and this interaction was hypothesized to promote reassembly of the 'XerC-active' synaptic complex into a conformation in which XerD is active and performs cleavage of the first DNA strand pair (*Aussel et al., 2002*; *Grainge et al., 2011*; *Zawadzki et al., 2013*). More recently, single-molecule Fluorescence Resonance Energy Transfer (smFRET) studies indicated that the majority of XerC/D-*dif* synaptic complexes formed in the absence of FtsK are in a conformation where XerD is catalytically inactive, but ready for direct activation by FtsK without major dis- and reassembly of the complex (*Zawadzki et al., 2013*). A small proportion of XerC active complexes nevertheless form and create unproductive HJs, but the pre-active XerD synaptic complexes are dominant and are the ones activated by FtsK. However, none of the previous structures or biochemical data have provided an explanation for why Xer recombinases require activation, or what the nature of the proposed conformational rearrangements might be.

To begin to shed light on these issues, we determined the structure of *Helicobacter pylori* XerH in complex with its $dif_H$ DNA substrate. $dif_H$ (*Figure 2A*) was previously predicted by comparative genome analysis and consists of two XerH-binding arms separated by a 6 bp central region (*Carnoy and Roten, 2009*). We confirmed XerH binding to $dif_H$ by DNase I footprinting (*Figure 2—figure supplement 1A*) and analytical size exclusion chromatography (SEC) (*Figure 2—figure supplement 1B*). We then co-crystallized full-length wild-type XerH with a 30 bp $dif_H$ DNA duplex. The resulting 2.1 Å resolution structure (*Figure 2B* and *Table 1*) shows a synaptic complex with two $dif_H$ molecules and four XerH subunits (*Figure 2B*, left panel). Each $dif_H$ site interacts with two XerH molecules, one molecule (molecule A) binding to the left arm and one (molecule B) to the right arm. The complex has overall two-fold symmetry relating the two DNA molecules and the XerH molecules bound to them (i.e. molecules A and B to A' and B'; *Figure 2B*). The overall fold of each protein subunit resembles that of other tyrosine recombinases, comprising two mostly helical domains

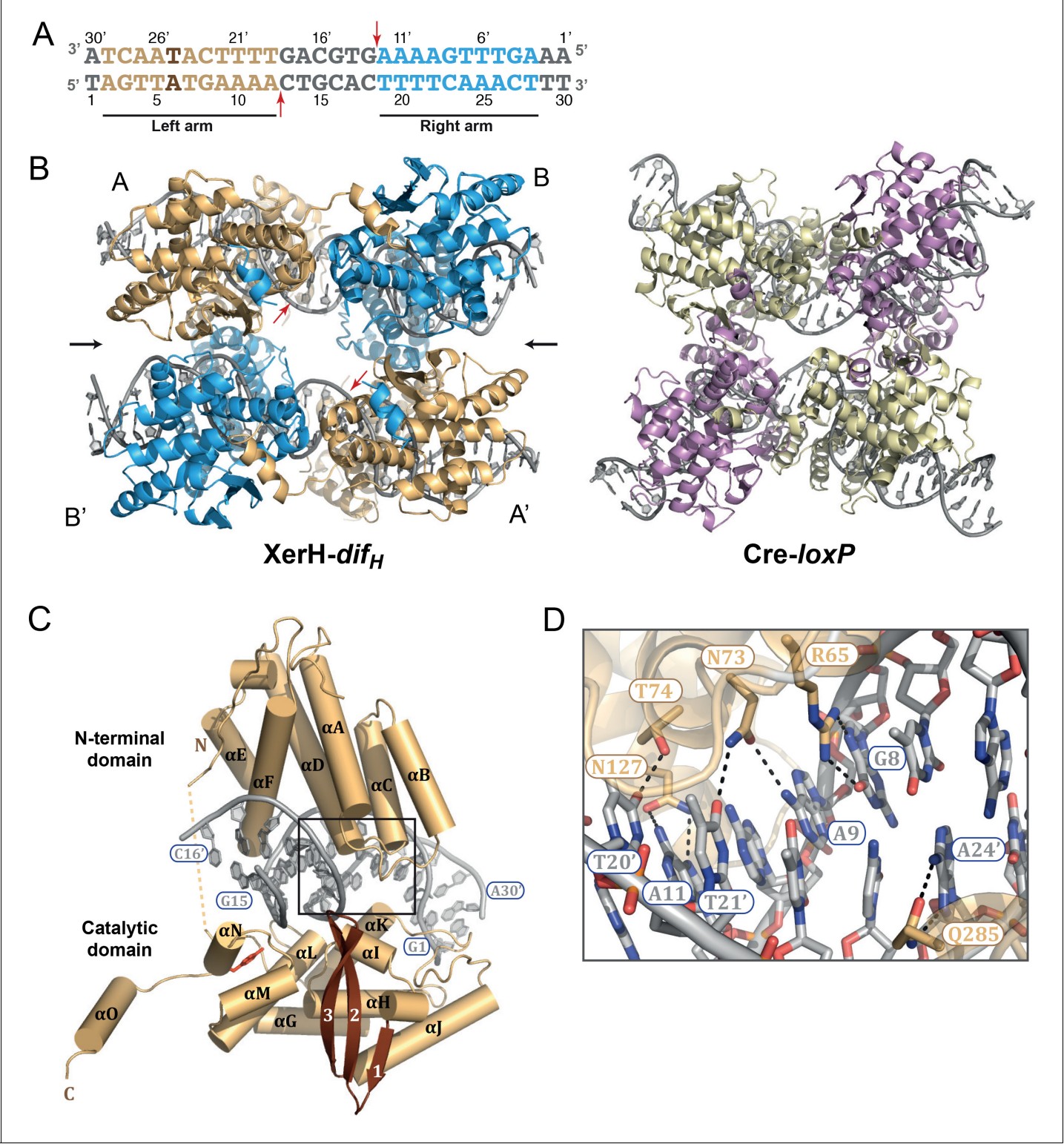

**Figure 2.** Structure of the XerH-*dif_H* complex. (**A**) Sequence of the *H. pylori dif_H* site. The sequence is written in the 3' to 5' direction to maintain consistency with the historical nomenclature of the arms, the structural figures, and the schematic models. The two XerH-binding arms are shown in gold and blue and the central region and the terminal base-pairs in gray. The inserted base-pair in the left arm is shown in brown. Red arrows indicate XerH cleavage sites. Binding assays confirming the site are shown in *Figure 2—figure supplement 1*. (**B**) The XerH-*dif_H* synaptic complex structure (left), compared to the previously solved Cre-*loxP* synaptic complex (PDB: 4CRX; right). XerH molecules are shown in cartoon representation, colored as their bound *dif_H* arms in A (molecule A in gold, molecule B in blue). Red arrows indicate the cleavage sites on the DNA. Side arrows mark

*Figure 2 continued on next page*

*Figure 2 continued*

the synaptic interface. The dyad symmetry axis of the tetramer runs along the midline of the synaptic interface. See *Figure 2—figure supplement 2* for a snapshot of the electron density map. (C) Close-up of the left arm-bound XerH (molecule A, gold). The N-terminal domain (residues 1–163) consists of six α-helices (αA to αF, cylinders); helices αB and αC are XerH-specific. The C-terminal catalytic domain (residues 171–362) is also mostly helical (αG-α O), with a single β-sheet containing three antiparallel β-strands (brown arrows). The interdomain linker could not be located in the structure (dashed line). The catalytic tyrosine is shown in red. *Figure 2—figure supplement 3* shows a comparison with the previously solved DNA-free structure of XerD. (D) Sequence-specific interactions of XerH (side chains shown as sticks with atomic coloring) and DNA. Hydrogen bonds (<3.5 Å) are shown as dashed black lines. See *Figure 2—figure supplement 4* for a comprehensive overview of the protein-DNA interactions and their biochemical validation.

The following figure supplements are available for figure 2:

**Figure supplement 1.** XerH binding to the predicted *dif_H* site.
**Figure supplement 2.** A cross-eyed stereo image of the bias-minimized 2Fo-Fc composite omit electron density map of the pre-cleavage XerH-*dif_H* complex structure.
**Figure supplement 3.** Comparison of the DNA-bound XerH (gold) and DNA-free XerD (PDB: 1A0P; green) structures.
**Figure supplement 4.** Interactions between XerH and the *dif_H* site.

connected by a 7-aa linker (*Figure 2C*). Notably, the C-terminal helix αO protrudes from the body of each catalytic domain into a cleft on the surface of an adjacent subunit in a cyclic fashion, creating a cyclic domain-swapped arrangement similar to the ones observed in the structures of Cre (*Figure 2B*, right panel) and λ integrase synaptic complexes (*Biswas et al., 2005*; *Guo et al., 1997*).

Whereas the structure of individual XerH subunits is similar to that of other tyrosine recombinases, the arrangement of the subunits within the synaptic complex differs from that observed previously, forming a considerably less compact synaptic interface with a wide open central channel (illustrated by comparison with the 'paradigm' Cre-*loxP* structure in *Figure 2B*) (*Guo et al., 1999*). Furthermore, the DNA molecules assume a near-straight conformation, in contrast to all previously reported tyrosine recombinase synaptic complexes.

## Recognition of the *dif_H* site

All four subunits of the XerH tetramer interact extensively with DNA. The two domains of each subunit form a tight, C-shaped clamp around one arm of the *dif_H* site (*Figure 2C*). The N-terminal domain contacts the DNA using a four-helix bundle (αA-αD, aa 12–88) and helix αF, both of which insert into the major groove (*Figure 2C*). These protein segments contribute most of the sequence-specific interactions to the DNA bases, while the catalytic domain contacts mainly the DNA backbone (*Figure 2—figure supplement 4A*). Many interactions of the catalytic domain involve helix αK (aa 285–299) that is inserted into the major groove at the outer part of *dif_H*, narrows the groove (17 Å as opposed to 22 Å in typical B-DNA) and induces a slight DNA bend (*Figure 2C*). In addition, several contacts dispersed along the DNA backbone help to stabilize the position of the catalytic domain.

XerH covers 11 bp of the left *dif_H* arm and 10 bp of the right arm (*Figure 2—figure supplement 4A*). The inner 5 bp of each arm (positions 7–11 and 20–24, *Figure 2A*) are recognized sequence-specifically (*Figure 2D*), while the outer nucleotides (positions 2–4 and 26–28) are mostly contacted at the phosphate backbone. The central region is only involved in sparse backbone interactions (*Figure 2—figure supplement 4A*). To confirm the importance of the *dif_H* DNA sequence, we performed cleavage assays with 'suicide' *dif_H* substrates (*Figure 2—figure supplement 4B*). These substrates contain a nick in the DNA backbone of each strand, one nucleotide downstream of the cleavage position. Upon XerH-mediated cleavage, the resulting covalent XerH-*dif_H* intermediate is trapped and can be detected by SDS-PAGE (see also *Figure 3C*). As predicted by the structure, mutations of the inner base-pairs of the arms (where XerH makes base-specific contacts) resulted in abolished or greatly reduced cleavage activity, whereas mutations of the neighboring base-pairs did not affect the activity.

**Table 1.** X-ray diffraction data collection and refinement statistics.

| | XerH-$dif_H$ | XerH-$dif_H$LP native | XerH-$dif_H$LP Se |
|---|---|---|---|
| **Crystal properties** | | | |
| Space group | P 2₁ 2₁ 2₁ | I 2 2 2 | I 2 2 2 |
| Unit cell: a, b, c (Å) | 79.28, 153.2, 169.39 | 86.38, 115.22, 235.2 | 85.79, 115.73, 235.29 |
| Unit cell: α, β, γ (°) | 90, 90, 90 | 90, 90, 90 | 90, 90, 90 |
| **Data collection** | | | |
| Beamline | I04-1 (DLS) | ID29 (ESRF) | ID29 (ESRF) |
| Wavelength (Å) | 0.92819 | 0.97908 | 0.97908 |
| Resolution range (Å) | 48.89–2.1 (2.18–2.1) | 46.96–2.4 (2.49–2.4) | 47.04–3.15 (3.2–3.15) |
| Total reflections | 793843 | 372682 | 100854 |
| Unique reflections | 120659 | 46216 | 37960 |
| Multiplicity | 6.6 (6.0) | 8.1 (8.2) | 2.7 (2.6) |
| Completeness (%) | 99.87 (99.81) | 99.86 (99.50) | 97.2 (96.2) |
| R-meas (%) | 11.28 (82.4) | 10.32 (140.1) | 11.9 (71.0) |
| R-sym (%) | 10.4 (75.4) | 9.7 (131.4) | 9.7 (58.2) |
| I/σI | 12.53 (2.25) | 14.90 (1.51) | 11.17 (1.89) |
| CC1/2 | 0.998 (0.754) | 0.999 (0.648) | 0.994 (0.727) |
| Wilson B-factor | 31.55 | 56.97 | 59.75 |
| | | | |
| **Refinement** | | | |
| R-work | 0.1913 | 0.1949 | |
| R-free | 0.2233 | 0.2203 | |
| Number of non-hydrogen atoms | 14670 | 6984 | |
| Protein residues | 1508 | 769 | |
| RMS (bonds) | 0.003 | 0.002 | |
| RMS (angles) | 0.53 | 0.55 | |
| Ramachandran favored (%) | 99 | 97 | |
| Ramachandran outliers (%) | 0 | 0 | |
| Clashscore | 1.92 | 2.99 | |
| Average B-factor | 37.90 | 72.20 | |

In contrast to the previously described DNA-free structure of XerD (*Subramanya et al., 1997*), the N-terminal and catalytic domains of XerH are positioned such that the cleft formed between their inner surfaces readily accommodates the $dif_H$ DNA in a conformation that resembles DNA-bound structures of other tyrosine recombinases (*Aihara et al., 2003*; *Chen et al., 2000*; *Guo et al., 1997*). A simple rotation of the N-terminal domain pivoting at the interdomain linker is sufficient to transition from the 'closed' XerD conformation to the 'open' DNA-bound XerH conformation (*Figure 2—figure supplement 3*), consistent with previous proposals that Xer recombinases may undergo a major conformational change upon DNA binding (*Jo et al., 2016*; *Serre et al., 2013*; *Subramanya et al., 1997*).

### $dif_H$ asymmetry dictates the order of binding and cleavage events

One of the hallmarks of tyrosine recombination is that the DNA strands are exchanged step-wise in a strictly ordered manner. In several studied examples, this ordering is believed to be linked to asymmetry in the DNA sequences, but the exact mechanisms vary (*Blakely et al., 1993*; *Ennifar et al., 2003*; *Nolivos et al., 2010*). To determine how the XerH-$dif_H$ system achieves

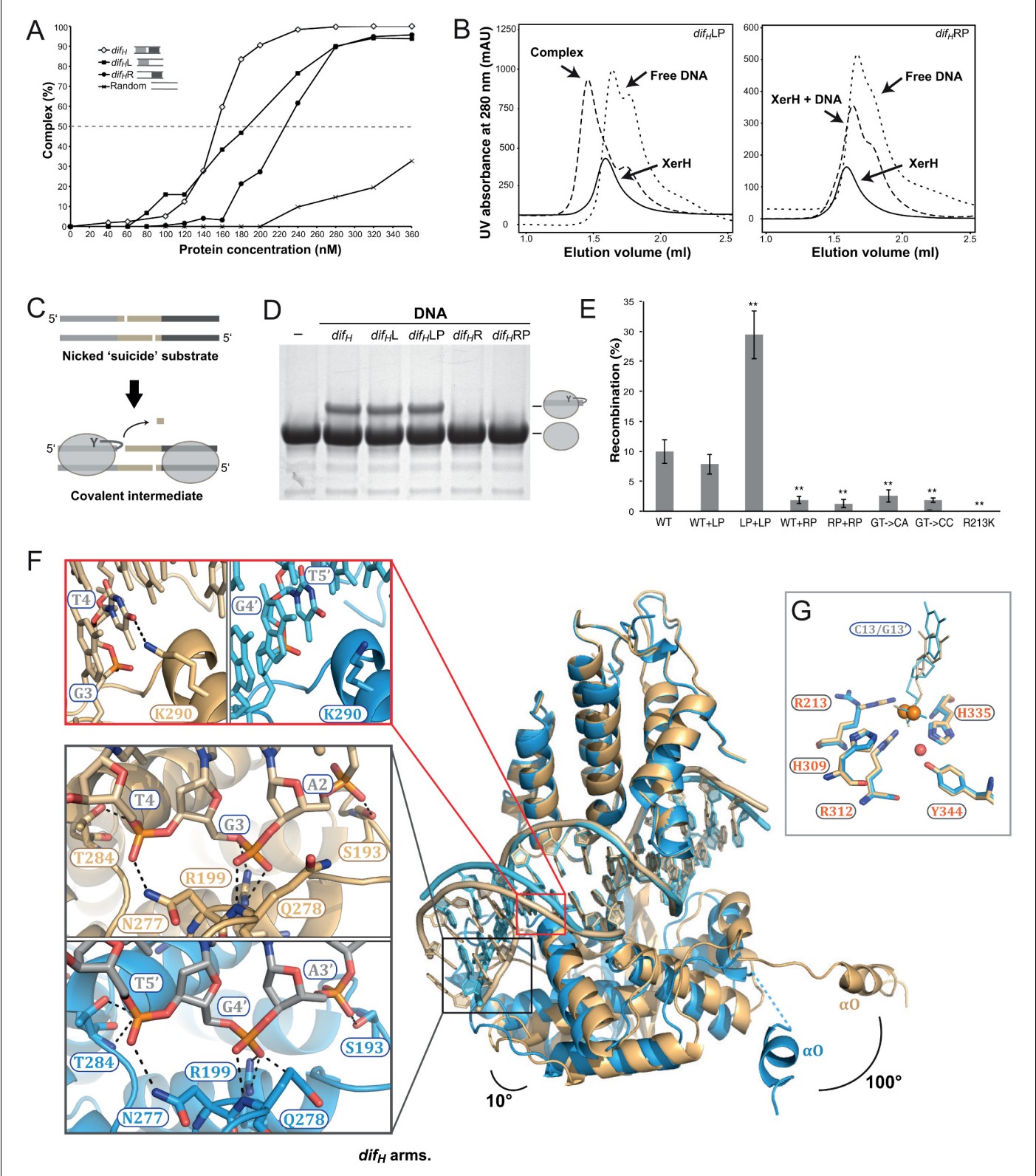

**Figure 3.** Differential recognition of the two $dif_H$ arms. (**A**) XerH binding to $dif_H$ and derivatives containing a single $dif_H$ arm ($dif_H$L or $dif_H$R) flanked by a random sequence, as quantified from EMSA (see original gels in ***Figure 3—figure supplement 1*** and numerical quantification data in ***Figure 3— source data 1***). Random DNA was used as a control. (**B**) SEC of XerH complexes with palindromic $dif_H$ substrates (left-arm palindrome $dif_H$LP, right-arm palindrome $dif_H$RP). XerH alone migrates as a monomer. A shift in retention is observed with $dif_H$LP, indicating stable complex formation. No

*Figure 3 continued on next page*

*Figure 3 continued*

corresponding shift is seen with $dif_H$RP. (**C**) The design of the in vitro cleavage assay using nicked 'suicide' substrates: upon cleavage by XerH, a single nucleotide diffuses away trapping the covalent phosphotyrosyl intermediate. (**D**) In vitro DNA cleavage assays, showing that the left arm of $dif_H$ is required for XerH activity. The covalent protein-DNA intermediate is detected by SDS-PAGE. (**E**) Left arm is required for XerH-mediated recombination in *E. coli*. Intramolecular recombination rates were measured with plasmids containing two $dif_H$ sites (WT+WT), one $dif_H$ site and one $dif_H$LP site (WT +LP), two $dif_H$LP sites (LP+LP), one $dif_H$ site and one $dif_H$RP site (WT+RP), two $dif_H$RP sites (RP+RP), two $dif_H$ sites with G3 and T4 mutated to CA, or two $dif_H$ sites with G3 and T4 mutated to CC. G3 was mutated together with T4, because it may interact with K290 and complement the role of T4. XerH catalytic mutant R213K XerH is shown as a control. Bars indicate standard deviation determined from three independent experiments (n = 3). **p<0.05 (Student's test). Colony counts and recombination rates are tabulated and their statistical analysis is shown in *Figure 3—source data 2*. (**F**) XerH recognizes the left (gold) and right (blue) $dif_H$ arms differently. Superimposition of two adjacent XerH subunits shows differences in the DNA bending, in the positions of the C-terminal domains, and in the protruding helices αO. Red insert: Interaction of K290 and T4 (dashed lines: hydrogen bonds) at the left arm (left panel) is absent at the right arm (right panel). Black insert: Interactions with the three outermost base-pairs of the left (top) and the right (bottom) $dif_H$ arms are remarkably similar despite the shifted DNA sequence. Functional characterization of these interactions and the role of the specific features of the left $dif_H$ arm are shown in *Figure 3—figure supplement 2*. (**G**) Active site conformations at the left (gold) and right (blue) $dif_H$ arm. Catalytic residues (sticks) are incompletely assembled around the scissile phosphates (orange spheres). The red sphere denotes a bound water molecule.

The following source data and figure supplements are available for figure 3:

**Source data 1.** Quantification of XerH binding to $dif_H$ variants based on EMSA experiments.
**Source data 2.** Results of the in vivo recombination assays.
**Figure supplement 1.** Electrophoretic mobility shift assay (EMSA) gels showing XerH binding to $dif_H$ and derivatives.
**Figure supplement 2.** Functional characterization of the role of the specific features of the left $dif_H$ arm.

ordering of the recombination steps, we first tested the roles of the two distinct protein-binding arms of $dif_H$ (*Figure 2A*) in XerH binding and activity. DNA binding assays (EMSA) with substrates containing the sequences of one or both arms (*Figure 3A* and *Figure 3—figure supplement 1*) showed that two XerH monomers bind cooperatively (see especially *Figure 3—figure supplement 1A*). The affinity of a $dif_H$L substrate containing only the left arm sequence, with a random sequence in place of the right arm, was higher (dissociation constant, $K_D$ = 180 nM; similar to wild-type $dif_H$ $K_D$ = 150 nM) than the affinity of a similar right arm-containing substrate $dif_H$R ($K_D$ = 230 nM). Similarly, a palindromic substrate containing two $dif_H$ left arms ($dif_H$LP) made a stable complex with XerH (observed by SEC), whereas the equivalent right arm substrate ($dif_H$RP) did not (*Figure 3B*). DNA cleavage assays also revealed different efficiencies at the two arms: no cleavage product was detected with $dif_H$R or $dif_H$RP, whereas $dif_H$L and $dif_H$LP substrates were cleaved as efficiently as the native $dif_H$ site (*Figure 3C and D*). An assay that assesses the efficiency of intramolecular recombination between two plasmid-borne sites in *E. coli* also indicated differential activity at the two $dif_H$ arms in vivo: the wild-type recombination rate (9.9% recombination-positive colonies) increased when both $dif_H$ sites were replaced with $dif_H$LP (29.4%; p=0.0016, t-test), but decreased to barely detectable levels with $dif_H$RP (1.3%; p=0.0021, t-test) (*Figure 3E*).

Our observation that XerH binds and preferentially cleaves the left arm of $dif_H$ led us to ask how the arms are distinguished. Remarkably, the sequences of the two arms are identical apart from a single base-pair insertion in the left arm (*Figure 2A*). In the crystal structure, the respective XerH subunits (A and B) make very similar interactions with the two $dif_H$ arms (*Figure 2—figure supplement 4A*). The inserted base-pair (A6/T25', brown in *Figure 2A*) is not recognized directly. Instead, the main difference in the recognition of the two arms involves a conserved thymine base (T4/T5') in the outer parts of the arms: In the left arm, T4 forms hydrophobic contacts with XerH αK and makes a specific hydrogen bond with lysine K290 (*Figure 3F*, red insert), whereas in the right arm T5' is closer to the center of $dif_H$ and cannot make these interactions. Surprisingly, the three outermost base-pairs of each $dif_H$ arm make very similar backbone interactions with their respective XerH subunits, despite the fact that they are shifted in space due to the insertion in the left arm (altering the 'helical phase' of the sequence by ~35°; *Figure 3F*, black insert; *Figure 2—figure supplement 4A*). This is possible because the conformation of the DNA is asymmetric – the left arm is more bent than

the right arm – and the catalytic domains of the protein subunits are rotated by ~10° relative to each other (*Figure 3F*). Together, the observed differences result in stronger interaction of XerH with the left *dif*$_H$ arm (ΔG = −21.3 kcal/mol, estimated by PISA [*Krissinel and Henrick, 2007*]) than with the right arm (ΔG = −15.4 kcal/mol). A further difference between the XerH subunits bound to the left and right arms of *dif*$_H$ concerns the C-terminal helices. While the αN-αO segment of molecule A is fully ordered, the linker and αO are partially disordered in molecule B. Also, helices αO point in different directions (~100° rotation; *Figure 3F*).

In agreement with the structure, *dif*$_H$ variants containing mutations at the inserted base-pair showed no or only moderate decreases in cleavage in vitro (*Figure 3—figure supplement 2A*) and recombination in vivo (*Figure 3—figure supplement 2B*). In contrast, substitutions of T4 practically abolished recombination in *E. coli* (*Figure 3E*) and in *H. pylori* (*Debowski et al., 2012a*); and mutation of K290 (K290S) decreased recombination activity by about half (*Figure 3—figure supplement 2D*). Insertion of an additional base-pair next to A6/T25' in the left arm, further shifting the positions of the three outermost base-pairs, also abolished cleavage completely (*Figure 3—figure supplement 2A*).

Like the arms, the central region of *dif*$_H$ is also asymmetric, with different nucleotides flanking the left and right cleavage sites (positions C13 and G13'). However, XerH does not contact these nucleotides in our structure, and cleavage assays with *dif*$_H$ variants carrying mutations at these positions revealed no reduction in activity (*Figure 2—figure supplement 4B*), suggesting that the identity of these nucleotides does not contribute to XerH binding and cleavage asymmetry.

Together, these data demonstrate that XerH binds preferentially to the left arm of *dif*$_H$ thanks to favorable interactions with its outer sequence (including T4). Asymmetric interactions with the two arms of *dif*$_H$ also appear to dictate distinct protein conformations in the synaptic complex, including differential positioning of the αN-αO segment that carries the nucleophilic tyrosine, perhaps helping to define which arm is cleaved first.

## A catalytically inactive, pre-cleavage synaptic complex, with almost straight DNA

Perhaps the most unexpected feature of the XerH-*dif*$_H$ synaptic complex structure is the DNA conformation, which is nearly straight. Both DNA molecules in the synapse resemble B-form DNA, with a wide angle (165°) between the two *dif*$_H$ arms (*Figure 2B*, left panel). This is in sharp contrast to other currently available DNA-bound structures of tyrosine recombinases, all of which contain strongly bent DNA. For example, the *loxP* DNA is bent to ~100° in the analogous pre-cleavage structures of Cre-*loxP* complexes (*Figure 2B*, right panel). This bent DNA conformation is maintained throughout the recombination reaction, allowing easy transition between the reaction intermediates (*Ennifar et al., 2003*; *Gopaul et al., 1998*; *Guo et al., 1999*) (see also *Figure 1B*).

In the active site of each XerH subunit, several conserved residues including the catalytic tyrosine Y344, two arginine (R213 and R312), and two histidine (H309 and H335) residues are assembled around the scissile phosphate, together forming a catalytic pocket characteristic of tyrosine recombinases. The active sites of the subunits are very similar, except for the R213 side chain which points in different directions in molecules A and B (*Figure 3G*). Electron density for another essential catalytic residue, K239, could not be observed in either of the subunits, so we presume that it is disordered. The catalytic tyrosine Y344 is far away (5.2–5.8 Å) from the scissile phosphate in all subunits, so the structure represents a catalytically inactive synaptic complex, implying a conformational change is required prior to catalysis.

## Conformational rearrangements in the post-cleavage synaptic complex

To capture XerH in a post-cleavage synaptic complex, we used 'suicide' *dif*$_H$ substrates (containing a nick in each DNA strand, as in the in vitro cleavage assays described above, *Figure 3C*). Attempts to co-crystallize XerH with suicide versions of the native *dif*$_H$ sequence were unsuccessful, but palindromic *dif*$_H$LP suicide substrates gave us crystals that diffracted to 2.4 Å (*Table 1*).

The resulting structure (*Figure 4A*) differs considerably from the pre-cleavage structure. Each *dif*$_H$LP site interacts with two XerH molecules in a tetrameric synaptic complex, but in this structure, the two halves of the tetramer are related by crystallographic two-fold symmetry (relating molecule A to A' and B to B'; *Figure 4A*). While the structures of the individual XerH subunits and their

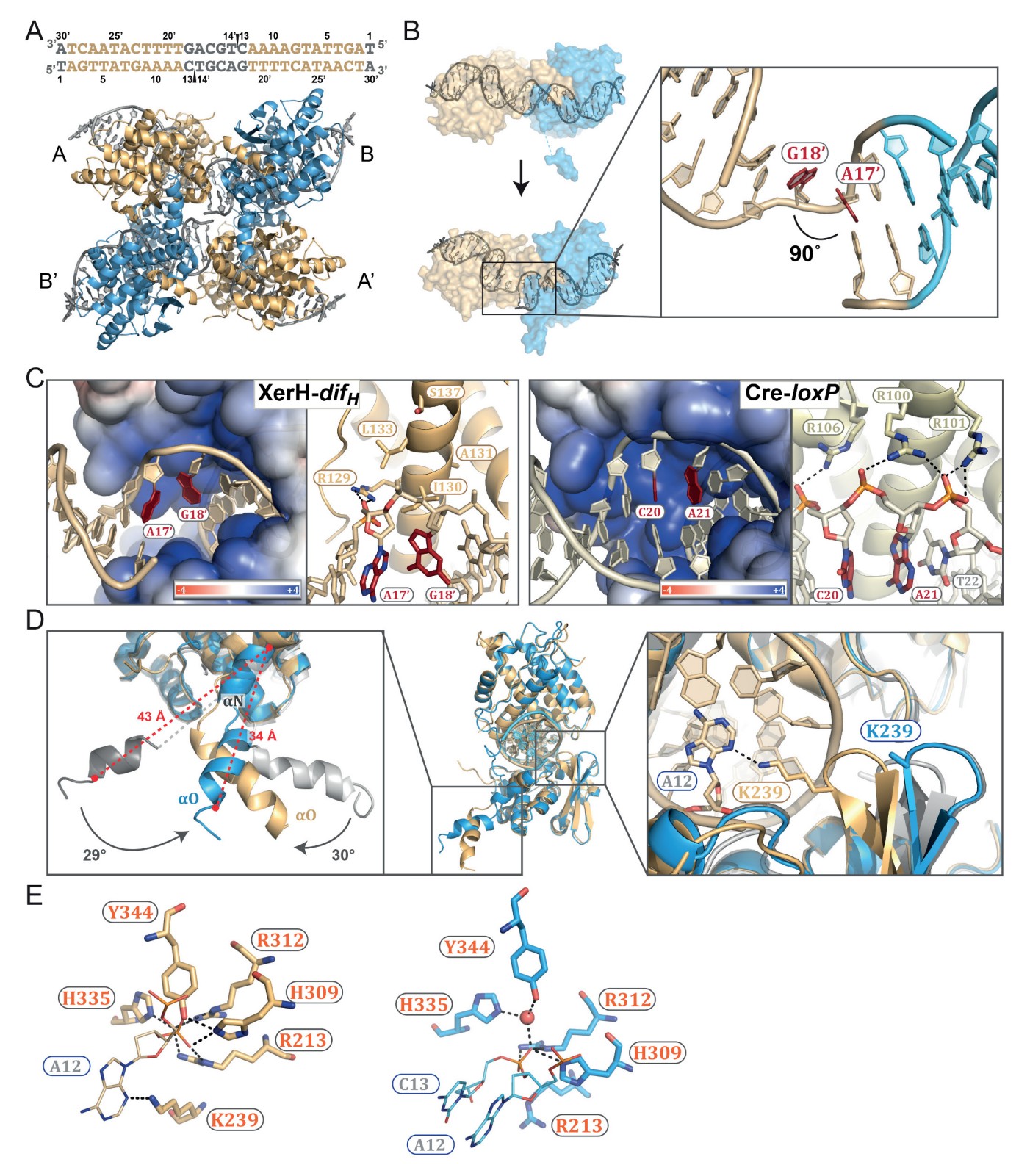

**Figure 4.** The post-cleavage XerH-*dif*$_H$LP synaptic complex structure. (**A**) Overall view of the XerH-*dif*$_H$LP structure in cartoon representation. Sequence of the *dif*$_H$LP substrate is shown above (written in the 3' to 5' direction, with the arms in gold, central region in gray) the nick positions are marked by triangles. *Figure 4—figure supplement 1* shows a snapshot of the electron density map. (**B**) Compared to the pre-cleavage XerH-*dif*$_H$ complex, both subunits (A, gold; B, blue surface) have rotated ~22° towards each other, concomitant with DNA bending. Insert: Close-up of DNA bending: A17' and

*Figure 4 continued on next page*

*Figure 4 continued*

G18' (red) are unstacked, with a 90° kink. (C) XerH (left) interacts with the DNA kink differently than Cre (PDB: 1NZB; right). The electrostatic surface potential is shown in red (negative) and blue (positive). (D) Superimposition of the two subunits (A, gold and B, blue) bound to the same *dif*$_H$LP site illustrates their different conformations. Left insert: Helices αN and αO are repositioned relative to the pre-cleavage structure (grey), including rotations (marked by arrows) and shortening of the helices (red dashed lines showing the distances between Cα of L338 and Cα of W359). Right insert: repositioning of the *β*2-*β*3 loop upon activation enables the catalytic K239 to interact with the DNA. For clarity, only the DNA associated with the golden monomer is shown. (E) Active site conformations of the distinct XerH subunits. In the active subunit (molecule A, left), R213, H309, R312, and H335 make hydrogen bonds with the scissile phosphate (orange), K239 contacts the base of the adjacent nucleotide A12, and Y344 is covalently attached to the DNA. In the inactive subunit (right) the catalytic tyrosine is 5.5 Å away from the scissile phosphate, R213 and H335 point away, and the K239 side chain is disordered. Red sphere – bound water molecule; dashed lines – hydrogen bonds.

The following figure supplement is available for figure 4:

**Figure supplement 1.** Cross-eyed stereo image of the bias-minimized 2Fo-Fc composite omit electron density map of the post-cleavage XerH-*dif*$_H$ complex structure.

interactions with the respective *dif*$_H$ DNA arms in this structure and in the pre-cleavage structure are similar, the overall arrangement of the complex is different. Most strikingly, the *dif*$_H$ DNA is now strongly bent (*Figure 4B*, *Video 1*). The bend mainly originates from a single distortion within the central region of *dif*$_H$: bases A17' and G18' are un-stacked with a 90° tilt, which introduces a kink resulting in asymmetric bending of the DNA (insert in *Figure 4B*). The total angle of 120° between the *dif*$_H$ arms is only slightly wider than the angles observed in previous structures of various tyrosine recombinases (e.g. 109° in the post-cleavage structure of λ integrase (*Biswas et al., 2005*) and 100° in Cre-*loxP* (*Guo et al., 1997*); see also *Figure 2B*). Notably, the kink in *dif*$_H$ is at an equivalent position to the kink in *loxP* (*Ennifar et al., 2003*). However, whereas in Cre the nucleotides involved in the kink interact with three arginines within a tight pocket (*Figure 4C*) (*Ennifar et al., 2003*; *Guo et al., 1997*), XerH forms only a single interaction with these nucleotides (R129-A17'; *Figure 4C*; *Figure 2—figure supplement 4A*) and encircles the kink less tightly. Notably, the nucleotides of the central region of *dif*$_H$ remain fully base-paired after cleavage, unlike the analogous structure of Cre-*loxP*, where several of these are unpaired (*Guo et al., 1997*). The extensive Cre contacts can thus actively bend *loxP*. In contrast, weaker XerH-*dif*$_H$ interactions appear to be insufficient to promote sharp DNA bending, but nicks in both DNA strands of the *dif*$_H$ substrate facilitate bending.

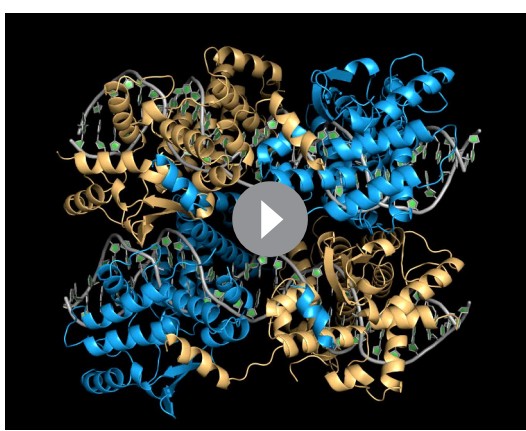

**Video 1.** Activation of the XerH-*dif*$_H$ synaptic complex. Morphing of the XerH-*dif*$_H$ synaptic complex in pre-cleavage conformation into the post-cleavage conformation. Activation involves major conformational rearrangement of the synaptic complex, involving rotation of all XerH subunits and sharp DNA bending.

When we tried to re-build the DNA strands in the post-cleavage structure to model a bent but un-nicked substrate, we observed a clash with the protein near the tyrosine nucleophile-bearing helix αN, suggesting that the nicks might favor the XerH conformation adopted on the bent sites.

The arrangement of the XerH subunits is also markedly different in the two structures. Each subunit is rotated by ~22° in the post-cleavage structure (*Figure 4B*, *Video 1*), resulting in major rewiring of the inter-subunit interactions and an overall compaction of the complex. Also, the synaptic interface is considerably larger (3926 Å², calculated by PISA [*Krissinel and Henrick, 2007*]) than in the pre-cleavage structure (3040 Å²). The previously disordered parts of the αO helices of molecule B are now fully visible, ordered, and more compact (F338 - W359 distance reduced from 43 Å to 34 Å; *Figure 4D*, left insert). At the same time, helices αN and αO of molecule A are rotated by ~30°, bringing the catalytic Y344 to a suitable position to attack

one strand on each *dif*<sub>H</sub>LP DNA forming a covalent phosphotyrosyl bond (*Figure 4E*, left).

Interestingly, despite the symmetry of the palindromic *dif*$_H$LP substrate, the two bound XerH molecules (A and B) are in different conformations (*Figure 4D*). Their overall structures are similar (r.m.s. deviation 0.78 Å for Cα carbons), but important protein segments that contain catalytic residues (the $\beta$2-$\beta$3 turn with K239, and αN-αO carrying Y344) are in different conformations (*Figure 4D*). Consequently, subunit A assumes a fully active conformation, while subunit B is in an inactive conformation with the catalytic tyrosine distant from the scissile phosphate (*Figure 4E*). This simultaneous presence of both active and inactive XerH subunit conformations within the synaptic complex is consistent with the 'half-of-the-sites reactivity' mechanism observed for other tyrosine recombinases (*Figure 1B*). The marked difference between the XerH subunits bound to the two *dif*$_H$LP arms (comparable to the asymmetry seen in the pre-cleavage complex with native *dif*$_H$) is particularly striking considering that the artificially introduced symmetry of the substrate could be expected to mask some asymmetric features of the synaptic complex. The asymmetry is presumably an intrinsic property of the system, essential to the mechanism of synapsis and catalysis.

## Modeling of *E. coli* XerC/D synaptic complex based on the XerH-*dif*$_H$ structures

Due to high structural conservation within the Xer family (*Subramanya et al., 1997*) (*Figure 5—figure supplement 1A*), our XerH structures can provide insights into the mechanisms of other Xer recombinases that have eluded structural studies so far. We have used our structures to model the heterotetrameric synaptic complex of *E. coli* XerC/D in both pre- and post-cleavage states (*Figure 5A*). We modeled DNA-bound XerC and XerD using the DNA-free XerD structure (PDB: 1A0P; (*Subramanya et al., 1997*)) and a homology model for XerC. These were then superimposed onto the XerH-*dif*$_H$ structures to assemble heterotetramers. Our models place XerC on the right arm of *dif*$_H$ and XerD on the left arm. We cannot exclude the alternative assignment, but our observation that XerH preferentially cleaves at the left arm of *dif*$_H$ first (*Figure 3D*) supports the hypothesis that the XerH subunit on the left arm corresponds to XerD whereas the subunit on the right arm corresponds to XerC.

In the models, the XerC and XerD monomers each form C-shaped clamps around one arm of the *dif* site (*Figure 5B*), and make multiple sequence-specific and backbone interactions with the DNA. The overall fold and conformation of the two proteins are very similar; only their C-terminal helices assume different relative orientations, as is also seen in the XerH structures. The five conserved catalytic residues (RHRHY) assemble around the catalytic pocket in all XerC and XerD subunits (*Figure 5C*). In the post-cleavage complex, the XerD subunits are in an active conformation, with their nucleophilic tyrosines approaching the scissile phosphates, whereas the XerC subunits are in an inactive conformation. As in the XerH structures, the three outer base-pairs of each *dif* arm are contacted extensively by the proteins (*Figure 5—figure supplement 1B*), and the bases corresponding to T4 in *dif*$_H$ interact with lysine K222 of XerC or histidine H225 of XerD, both counterparts of XerH K290 (*Figure 5—figure supplement 1B*). These interactions might contribute to the arm specificity of XerC and XerD, as well as to ordering recombination, as we inferred for XerH. The interactions involved in DNA kinking in Cre are absent in the XerC/D-*dif* complex, suggesting that these enzymes, like XerH, might be unable to independently initiate sharp DNA bending.

The protein arrangements and interfaces observed in our two XerH-*dif*$_H$ structures are well accommodated by XerC and XerD, the pre-cleavage model showing nearly straight DNA and the post-cleavage model containing bent DNA and tightened intersubunit interactions (*Figure 5A*). This suggests that the XerC/D and XerH complexes might undergo similar conformational rearrangements upon activation. In agreement with this idea, two previous smFRET studies (*Diagne et al., 2014*; *Zawadzki et al., 2013*) reported that pre-formed XerC/D-*dif* synaptic complexes undergo conformational change upon activation by FtsK, leading to an increase of the distance between the two *dif* termini from about 53 to 67 Å across the synapse (as predicted from a change in FRET efficiencies; [*Zawadzki et al., 2013*]). Our XerC/D-*dif* models predict a similar change from 46 Å in the pre-cleavage state to 60 Å in the post-cleavage state.

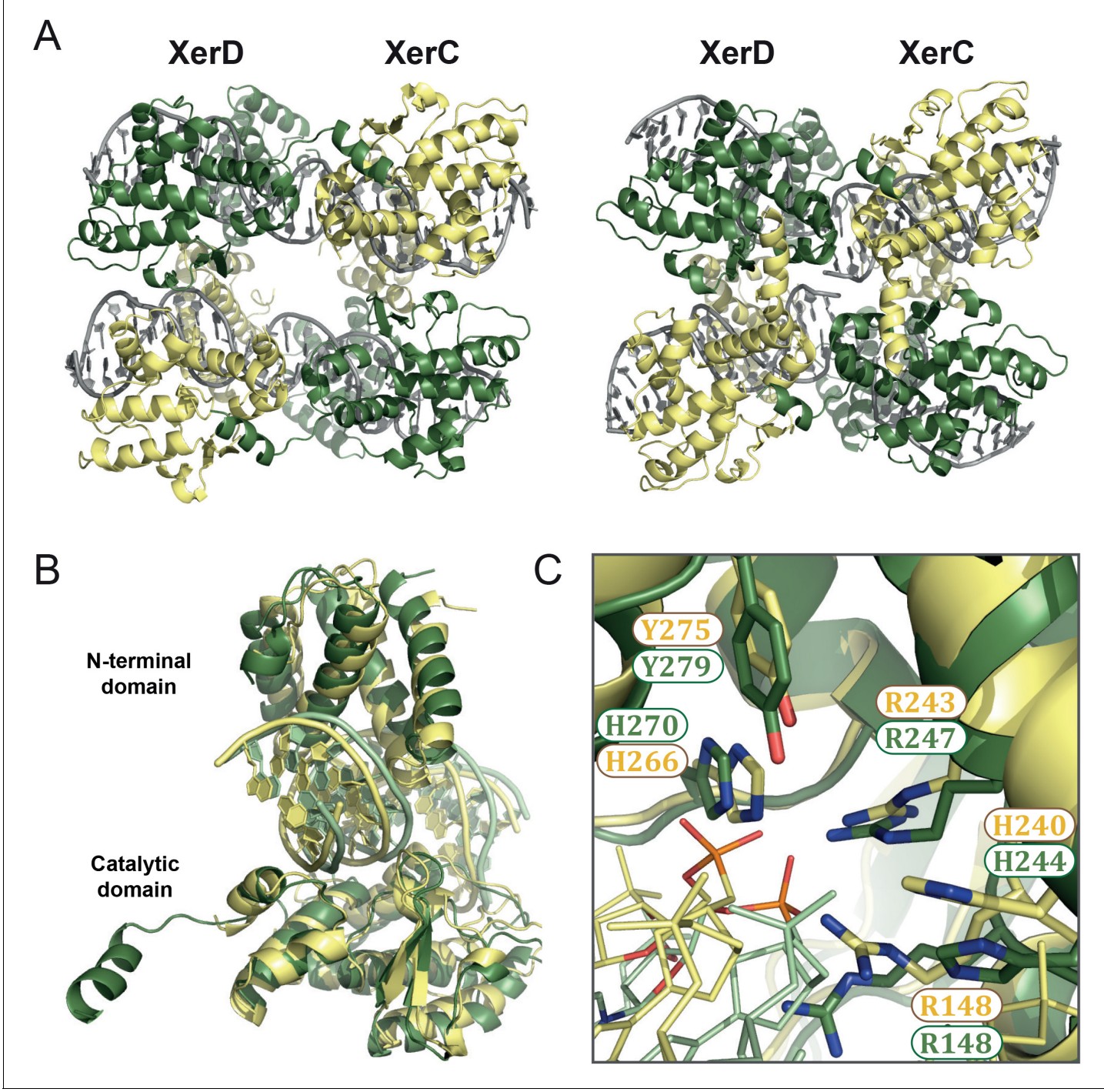

**Figure 5.** XerC/D-*dif* synaptic complex models reveal common features of Xer recombination. (**A**) Cartoon representation of heterotetrameric XerC/D-*dif* synaptic complexes modeled based on the pre-cleavage (left) or the post-cleavage (right) XerH-*dif*$_H$ complexes. XerD (green) and XerC (yellow) monomers are arranged compatibly in the tetramers. *Figure 5—figure supplement 1* shows the structure-based sequence alignment and a comparison of protein-DNA interactions. (**B**) Superimposition of XerD and XerC monomers in the pre-cleavage structure model. (**C**) Conserved active site residues of XerD and XerC (sticks with atomic coloring) are assembled around the scissile phosphate (orange) in the pre-cleavage XerC/D-*dif* synaptic complex.

The following figure supplement is available for figure 5:

**Figure supplement 1.** Modeling of XerC/D-*dif* synaptic complexes.

## Discussion

In this work, we investigated the structural and mechanistic bases of Xer site-specific recombination. Using the homomeric XerH-$dif_H$ system from *H. pylori* as a model system, we report the first high-resolution crystal structures of Xer-*dif* synaptic complexes. These structures demonstrate that Xer proteins follow the established chemical pathway of tyrosine recombinases and reveal how the reaction steps can be choreographed by small differences in the arms of the $dif_H$ recombination sites. Together with associated biochemical data, the structures also show that XerH-$dif_H$ synaptic complexes initially assemble in an inactive state with straight DNA, which must undergo a major conformational change for catalytic activation, as previously observed in single molecule fluorescence studies of XerC/D-*dif* recombination (*Diagne et al., 2014*; *Zawadzki et al., 2013*). Our post-cleavage XerH-$dif_H$ synaptic complex structure elucidates the structural nature of this conformational change, which involves major rearrangement of the protein-protein interfaces and DNA bending (*Video 1*). With molecular modeling, we extend our structural and mechanistic findings to the prototypical *E. coli* XerC/D-*dif* system, demonstrating that our structures provide a resource for understanding the mechanism of other Xer recombinases.

The thoroughly characterized Cre site-specific recombination reaction has long been considered a paradigm for tyrosine recombinase-based DNA rearrangements (*Gopaul and Duyne, 1999*; *Grindley et al., 2006*; *Guo et al., 1997*; *Van Duyne, 2015*). Unusually, Xer recombinases require an external factor (generally the cell division protein FtsK) for catalytic activation. This feature is essential for Xer's chromosome maintenance function, but does not have any analogy in the Cre system. Therefore, the structural basis of Xer recombination and its activation have remained debated. From our structural, biochemical, and microbiological data on *H. pylori* XerH-$dif_H$ recombination we can

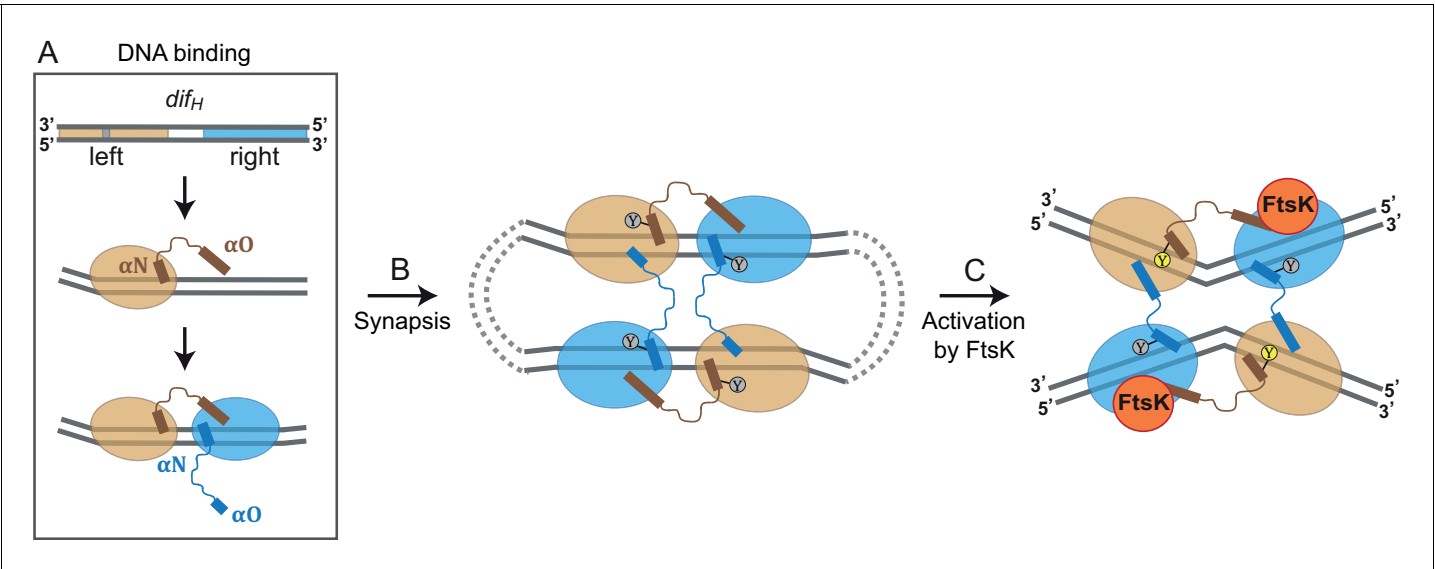

**Figure 6.** Model for XerH recombination activation. (A–C) Differential binding affinities of XerH to the left and right arms of $dif_H$ DNA trigger an asymmetric DNA conformation and arrange the two protein subunits in distinct conformations (A). Two $dif_H$ sites are brought together in a tetrameric synaptic complex (B) stabilized by a cyclic arrangement of the protruding αO helices. Initially formed XerH-$dif_H$ complexes are inactive (indicated by gray nucleophilic tyrosine) as seen in our pre-cleavage structure. *Figure 6—figure supplement 1* shows that XerH alone is unable to cleave an intact DNA substrate. Catalytic activation then involves DNA bending and a major rearrangement of the protein subunits (as in our post-cleavage structure), presumably enabled by a direct interaction with FtsK (C). As a result, helices αO become ordered across the synapse, while helices αN and αO along one $dif_H$ molecule rotate, bringing the nucleophilic Y344 into the active conformation (yellow) that enables DNA cleavage. Note that the DNA substrates are drawn with the strand going 3' to 5' on the top as in *Figure 1B*.

The following figure supplement is available for figure 6:

**Figure supplement 1.** XerH activity on intact and nicked $dif_H$ substrates.

now propose a mechanistic model for XerH recombination and the regulatory function of FtsK, as follows (*Figure 6*):

The recombination process starts with XerH binding to the *dif_H* site on the *H. pylori* chromosome (*Figure 6A*). Two XerH subunits bind cooperatively to one *dif_H* site (*Figure 3—figure supplement 1*; *Figure 6A*), each forming a clamp around one arm of *dif_H* (*Figure 2C*). The *dif_H* left arm has higher affinity for XerH than the right arm (*Figure 3A,B*), leading to the second XerH molecule binding to the right arm more often than vice versa. Cooperative binding of XerH involves extensive contacts between the two subunits bound to each *dif_H*, as revealed by our structures (1650.6 Å$^2$ surface area with $\Delta$G = −12.6 kcal/mol and 1515.5 Å$^2$ with $\Delta$G = −14.5 kcal/mol between molecules A-B and A'-B' respectively in the pre-cleavage structure; and 1379.8 Å$^2$ surface area with $\Delta$G of −13.8 kcal/mol in the post-cleavage structure). Despite the near-perfect dyad symmetry of the *dif_H* arms, the bound XerH subunits are structurally distinct. The left arm subunit assumes a conformation primed for DNA cleavage, whereas the right arm subunit is forced in an inactive state. This asymmetric configuration defines the order of all subsequent cleavage and strand transfer events.

Two XerH-bound *dif_H* sites then interact to create a tetrameric synaptic complex, as we see in our crystal structures (*Figure 2B*; *Figure 6B*). The sites align in antiparallel, and intersubunit interactions across the synaptic interface anchor the αO helices of the right arm-bound XerH subunits on their left arm-bound partner, creating a 'circular' domain-swapped arrangement, similar to the ones described for other tyrosine recombinases such as Cre and λ integrase (*Biswas et al., 2005*; *Guo et al., 1997*).

The pre-cleavage synaptic complex crystal structure contains nearly straight *dif_H* DNA (*Figure 2B*; *Figure 6B*), and XerH is in a catalytically inactive conformation (*Figure 3G*). Correspondingly, we were unable to observe *dif_H* recombination or XerH-mediated cleavage of intact *dif_H* DNA substrates in vitro (*Figure 6—figure supplement 1*, and data not shown). We cannot exclude the possibility that XerH does still catalyze transient strand cleavage of these substrates, followed by rapid efficient re-ligation making cleavage undetectable, but there is no evidence to support this idea. In contrast, nicked *dif_H* suicide substrates are cleaved efficiently, and our crystal structure in which XerH-mediated cleavage of nicked *dif_H* has occurred showed a sharply bent DNA conformation, consistent with a direct link between DNA bending and cleavage activity. The crystals giving our two structures were grown under different conditions (see Materials and methods), but XerH was active (on nicked *dif_H* substrates) in both conditions (data not shown), supporting our view that both structures are biologically relevant. We hypothesize that our in vitro systems lack a stimulatory factor that is required for normal recombination between intact *dif_H* sites. Septum-borne FtsK was shown to be required for XerH recombination in *H. pylori* (*Debowski et al., 2012a*), so we propose that this is the missing factor for XerH activation in vitro. FtsK-promoted rearrangement of the XerH-*dif_H* synaptic complex might bring the catalytic tyrosines of the left arm-bound subunits into their active positions close to the scissile phosphates and sharply bend the DNA as seen in our structures (*Figures 4B* and *6C*; *Video 1*). Another possibility is that FtsK mediates formation of a different synaptic complex with the right arm-bound subunits activated for cleavage. However, this would be inconsistent with smFRET studies of XerC/D complexes showing that FtsK activates pre-formed synaptic complexes without major remodeling (*Zawadzki et al., 2013*). Analogy with the XerC/D system suggests that FtsK might directly interact with the C-terminal ~20 amino acids of XerH (*Yates et al., 2006*) or the back of its C-terminal domain (*Keller et al., 2016*). However, the role of FtsK in the *H. pylori* Xer system still requires further substantial investigation, and it remains possible that the bent-DNA configuration required for DNA cleavage can be reached from the pre-cleavage structure without the intervention of FtsK.

Following activation, XerH-catalysed DNA strand cleavage and rejoining presumably follow the conserved tyrosine recombination pathway (*Figure 1B*), with the XerH subunits bound to the left arms performing the first strand exchanges. The resulting HJ is then resolved by a reciprocal set of chemical reactions catalysed by the XerH subunits bound to the right arms.

DNA bending is a prerequisite for activity of many molecular machines, including transcription factors, topoisomerases and recombinases (*Dong and Berger, 2007*; *Kim et al., 1993*; *Lee et al., 2013*; *Yang and Steitz, 1995*). In most of these cases, DNA bending and the downstream function are performed by the same protein or complex, or DNA bending depends on ubiquitously available factors such as IHF. For tyrosine recombinases, the paradigm suggests that sharp DNA bending that is required for DNA cleavage and energetically inexpensive strand exchange, is introduced by the

recombinase itself concomitant with DNA binding and synapsis. Here, we show that this is not the case for XerH, which binds and assembles its recombination substrates in an almost straight conformation. The required DNA bending then presumably occurs upon activation by an external factor that is present only in a particular spatial and temporal context, providing an elegant regulatory mechanism to ensure faithful chromosome segregation.

Our modeling of XerC/D-*dif* indicates that the architectures of the pre- and post- cleavage synaptic complexes are probably conserved between XerC/D and XerH (*Figure 5B*). The DNA conformational changes that we observe for XerH-*dif*$_H$ are also consistent with smFRET studies of XerC/D-*dif* synaptic complexes (*Diagne et al., 2014*; *Zawadzki et al., 2013*), which imply a conformational change upon FtsK-mediated activation consistent with the differences between our models of the pre-cleavage and post-cleavage complexes. Thus, we expect that the mechanism of catalytic activation proposed here for XerH – preassembly of a synaptic complex with almost straight DNA, followed by FtsK-induced protein and DNA rearrangements activating the complex for catalysis – may be conserved in the XerC/D system.

In summary, our structures of XerH-*dif*$_H$ complexes demonstrate that catalysis by the Xer family of site-specific recombination systems follows the established tyrosine recombinase pathway, with the reaction steps being choreographed by small differences in the arms of the *dif* recombination sites. Contrary to previous assumption, bending of the *dif* sites does not occur concomitantly with synaptic complex assembly but at a post-synaptic step, when the accessory factor FtsK might be needed to license the reaction by promoting a conformational change. We also provide structural insight into the conformational change required for activation (*Video 1*), and extend our findings to other Xer recombinases through modeling. Our structural insights help us to better understand how Xer proteins function and how they have adapted the tyrosine recombinase machinery for their unique genome maintenance function. In the long term, our data can also help to improve XerH-based genetic engineering tools that have been recently introduced for markerless gene deletions in *H. pylori* (*Debowski et al., 2012b*).

# Materials and methods

## DNA constructs

Full-length *xerH* from *H. pylori* strain 26995 (NCBI: HP0675) was synthesized with codon-optimization (MrGene, Regensburg, Germany) for over-expression in *E. coli* and cloned into vector pETM-28 (PepCore, EMBL) using BamHI/XhoI restriction sites. XerH Y344F, R213K, S161A, and K290S constructs were prepared by site-directed mutagenesis using primers listed in *Supplementary file 1*.

## Protein expression and purification

*H. pylori* XerH and its mutants were overexpressed in *E. coli* BL21 (DE3) as N-terminal fusions with hexahistidine and Small Ubiquitin-like Modifier (SUMO) tags. The constructs were expressed in *E. coli* BL21 (DE3) for 16 h at 15°C after addition of 0.5 mM isopropyl $\beta$-D-1-thiogalactopyranoside (IPTG). Cells were lysed by sonication in purification buffer (1x phosphate buffered saline (PBS), 1 M NaCl, 5% glycerol, and 0.2 mM tris(2-carboxyethyl)phosphine (TCEP), pH 7.5) supplemented with protease inhibitors (cOmplete Protease Inhibitor Cocktail, Roche Diagnostics, Mannheim, Germany; 1.5 mM phenylmethanesulfonylfluoride [PMSF]). The lysate was cleared by centrifugation at 40,000 g. The protein was purified from the soluble fraction by Ni-affinity chromatography on a HisTrap column (GE Healthcare, Munich, Germany), followed by tag cleavage with SenP2 protease, tag removal on a Ni-affinity column, and size exclusion chromatography on Superdex 200 column (GE Heathcare). The seleno-methionine derivative of XerH was expressed in BL21 (DE3) in M9 growth medium supplemented with the essential amino acids, with seleno-methionine replacing methionine, and was purified as above.

## DNase I footprinting

The XerH-*dif*$_H$ complexes were formed in activity buffer (25 mM HEPES, pH 7.5, 100 mM NaCl, 10 mM MgCl2, 5% glycerol, 1 mM DTT, and 1 mM EDTA). 120 bp substrates containing the left or right TnPZ end were PCR-amplified from *H. pylori* 26695 genomic DNA using 5'-$^{32}$P-labelled primers shown in *Supplementary file 1*. Samples were incubated with various amounts of DNase I for 1 min

after addition of 5 mM $CaCl_2$ and 10 mM $MgCl_2$. The reactions were stopped with 120 mM NaCl, 18 mM EDTA, and 0.6% SDS, DNA was purified by ethanol precipitation and analyzed by PAGE on Urea-TBE 12% gel. Sequencing ladders were prepared with DNA Cycle Sequencing Kit (Jena Bioscience, Jena, Germany) using 5'-$^{32}$P-labeled primers.

## Crystallization and data collection

DNA oligonucleotides were synthesized and HPLC-purified by IDT (Leuven, Belgium), then annealed in TE buffer by heating to 98°C and slow cooling to room temperature or 4°C. XerH-$dif_H$ complexes were formed by mixing XerH with DNA (sequences shown in *Supplementary file 1*) at a 1.2:1 molar ratio in purification buffer, dialyzed in three steps to crystallization buffer containing 25 mM sodium acetate buffer (pH 5.5), 200 mM NaCl, 5% glycerol, and 1 mM DTT, and concentrated to 5 mg/ml. The crystallization conditions were screened using The Classics Suite (QIAGEN, Hilden, Germany), Index (Hampton Research, Aliso Viejo, CA, USA), and JCSG+ (*Page et al., 2003*) sparse matrix crystallization screens by sitting drop vapor diffusion. The resulting hits were scaled-up and optimized in hanging-drop vapor diffusion setup. 2 µl of complex solution were mixed with 2 µl of well solution and equilibrated against 0.5 ml well solution for two weeks at 6°C. The pre-cleavage complex was crystallized using well solutions containing 0.2 M NaCl and 20–25% (w/v) polyethylene glycol (PEG) 3350, while for the post-cleavage complex 0.1 M HEPES, pH 7–8, 0.2 M $MgCl_2$, and 25–35% (v/v) PEG 400 was used. For data collection, the crystals were harvested and flash-frozen in liquid nitrogen. The pre-cleavage complex crystals were cryoprotected by the addition of 20% v/v glycerol. The data were collected on beamline ID-29 at the European Synchrotron Radiation Facility (ESRF) and on beamline I04-1 at the Diamond Light Source (DLS).

## Structure determination

The diffraction data were processed with XDS (*Kabsch, 2010*). The structure of the post-cleavage complex was determined by SAD phasing in AutoSHARP (*Vonrhein et al., 2007*), using anomalous data from a seleno-methionine derivative crystal. The phases were extended onto a native dataset, and the initial model was built in AutoBuild in Phenix (*Adams et al., 2010*; *Terwilliger et al., 2008*). The asymmetric unit contained two XerH molecules and one nicked $dif_H$LP DNA, with two-fold crystallographic symmetry generating a tetramer.

The structure of the pre-cleavage complex was solved by molecular replacement in Phaser (*McCoy et al., 2007*), using an XerH monomer as a search model. This structure showed four XerH molecules and two $dif_H$ DNA per asymmetric unit. The final models were obtained by alternating model building in *Coot* (*Emsley et al., 2010*) and simulated annealing, restrained positional and B-factor refinement with Phenix (*Afonine et al., 2012*). The data collection and refinement statistics are given in *Table 1*. Protein interfaces were analyzed in PISA (*Krissinel and Henrick, 2007*), protein-DNA contacts in NUCPLOT (*Luscombe et al., 1997*), and DNA topology in W3DNA (*Zheng et al., 2009*). All structural figures were made in PyMOL (Version 1.5.0.4; Schrödinger, LLC, New York, NY, USA) and animations in Chimera (*Pettersen et al., 2004*).

## Binding assays

For electrophoretic mobility shift assays (EMSA), XerH was incubated with 50 bp $dif_H$ DNA or its derivatives (*Supplementary file 1*) in the activity buffer as for DNase I footprinting for 1 h at 30°C. 20 nM 5'-$^{32}$P-labelled DNA was titrated with increasing amounts of XerH (0–360 nM), and complex formation was assessed on native 12% polyacrylamide TBE gels. Gels were imaged in a Typhoon FLA 7000 Phosphoimager, and images were quantified and analyzed with ImageQuant (GE Healthcare). Dissociation constants ($K_D$) were calculated for the sum of all complex species. XerH-$dif_H$ complexes were also analyzed by analytical size exclusion chromatography on Superdex 200 3.2/30 column on Äkta systems (GE Healthcare). The XerH-DNA complexes were prepared as for crystallization.

## In vitro activity assays

Cleavage assays were carried out in activity buffer with 25 µM XerH and 25 µM $dif_H$ substrates. 'Suicide' $dif_H$ substrates containing nicks in the backbone of both DNA strands 1 nt 3' to the cleavage position were used. These substrates trap the covalent XerH-$dif_H$ reaction intermediates (*Figure 3C*)

as seen for Cre recombinase and λ integrase (*Guo et al., 1997*; *Pargellis et al., 1988*). Oligonucleotide sequences are shown in *Supplementary file 1*. DNA substrates were annealed by slow cooling from 98°C to 4°C. The reactions were incubated for 1 h at 37°C, and analyzed by SDS-PAGE.

### Recombination assay

XerH-mediated intramolecular recombination was assessed in *E. coli* by a *galK* marker-based assay, similar to the one described by (*Arnold et al., 1999*). The *dif$_H$* cassette was PCR-amplified from *H. pylori* 26695 genome using primers shown in *Supplementary file 1* and inserted in a direct repeat orientation at two positions of plasmid pMS183Δ using NheI/BsrGI or EcoRI/KpnI restriction sites. The mutated variants of the *dif$_H$* cassette were obtained by site-directed mutagenesis of the constructed plasmids using primers shown in *Supplementary file 1*. The *xerH* gene was cloned from the expression plasmids used for XerH production into plasmid pBAD/MCS (PepCore, EMBL) using NcoI/XhoI restriction sites. GalK-deficient *E. coli* strain DS941 was sequentially transformed first with the XerH expression plasmid, and then with the reporter plasmid. Transformants were plated on 4% (w/v) MacConkey agar (Difco$^{TM}$, Becton and Dickinson, Heidelberg, Germany) supplemented with 1% (w/v) galactose, 50 µg/ml kanamycin, and 100 µg/ml ampicillin. Upon XerH expression, recombination between the two *dif$_H$* sites leads to loss of *galK*. After overnight growth the plates were scraped to resuspend the cells in 1 ml of LB, and an overnight culture was set up with 1 µl of the scraped cells. *E. coli* DS941 cells were transformed with the plasmid DNA extracted from the overnight cultures, and the cells were plated again on MacConkey agar supplemented with 1% (w/v) galactose and 50 µg/ml kanamycin. Red (indicating no recombination) and white (indicating recombination deleting the *galK* gene from the plasmid) colonies were counted and recombination efficiency was calculated as the number of white colonies divided by the total number of colonies. Representative plasmids from white colonies were analyzed by agarose gel electrophoresis and sequencing to confirm that reciprocal recombination between *dif$_H$* sites had occurred as predicted.

### Modeling of XerC/D-*dif* complexes

First, DNA-bound XerC and XerD were modeled using the DNA-free XerD structure (PDB: 1 A0P) (*Subramanya et al., 1997*) and an XerC homology model created using I-TASSER (*Yang et al., 2015*). XerC/D tetramers were assembled by superposing individual protein domains onto our XerH-*dif$_H$* structures by rigid structural alignment using FATCAT (*Li et al., 2006*) with XerD placed onto the left *dif$_H$* arm (i.e. superposed onto subunit A in the cleavage-competent conformation). Flexible parts that could not be aligned (αM-αO and β2-β3 hairpin) were isolated and modeled by threading with Phyre2 (*Karaca and Bonvin, 2011*; *Kelley et al., 2015*). *dif* DNA was modeled with mode-RNA server (*Rother et al., 2011*) using *dif$_H$* as a template and refined in HADDOCK (*van Zundert et al., 2016*). Both pre- and post-cleavage complexes were modeled and the assembled models were refined in HADDOCK to optimize molecular geometry.

### Database accession codes

Coordinates and structure factors have been deposited in the Protein Data Bank under accession codes 5JK0 (XerH-*dif$_H$* pre-cleavage synaptic complex) and 5JJV (XerH-*dif$_H$*LP post-cleavage synaptic complex).

## Acknowledgements

This work was supported by: the EMBL; the EMBL International PhD Programme (fellowship to AB); Humboldt Postdoctoral Research Fellowship (to EK). We thank Drs. Sean Colloms, Teresa Carlomagno, Nassos Typas, and Francois-Xavier Barre for helpful discussions; Dr. Bianca Beusink and Pablo S Cortes for preparation of the initial constructs; Dr. Sebastian Glatt and the tutors of the DLS-CCP4 Data Collection and Structure Solution Workshop (2015) for help with data collection and processing; Drs. Anna Rubio Cosials, Alison B Hickman and Christoph W Müller for comments on the manuscript; Dr. Bernd Klaus from the EMBL Centre for Statistical Data Analysis for advise on statistical analysis; Dr. Cecilia Zuliani for technical assistance; the Protein Expression and Purification Core Facility and the Crystallization Facility at EMBL Heidelberg for materials and support. X-ray diffraction data were collected at ESRF (beamline ID29) and at DLS (beamline I04-1). We thank the staff of ESRF, EMBL-Grenoble and DLS for their assistance and support in using the beamlines.

## Additional information

### Funding

| Funder | Grant reference number | Author |
|---|---|---|
| European Molecular Biology Laboratory | Intramural Funds | Aleksandra Bebel<br>Ezgi Karaca<br>Banushree Kumar<br>Orsolya Barabas |
| Alexander von Humboldt-Stiftung | Postdoctoral Fellowship | Ezgi Karaca |
| European Molecular Biology Laboratory | International PhD Programme Graduate Student Fellowship | Aleksandra Bebel |

The funders had no role in study design, data collection and interpretation, or the decision to submit the work for publication.

### Author contributions

AB, Designed research and analyzed the results, Wrote the manuscript, Solved the crystal structures, Performed biochemical assays, Performed in vivo assays; EK, Performed structural analysis and modeling, Contributed to writing of the manuscript; BK, Performed in vivo assays ; WMS, Wrote the manuscript, Provided *E. coli* plasmids and strains; OB, Designed research and analyzed the results, Wrote the manuscript

### Author ORCIDs

Orsolya Barabas, http://orcid.org/0000-0002-2873-5872

## Additional files

### Supplementary files

• Supplementary file 1. List of oligonucleotides used in this study.

### Major datasets

The following datasets were generated:

| Author(s) | Year | Dataset title | Dataset URL | Database, license, and accessibility information |
|---|---|---|---|---|
| Bebel A, Barabas O | 2016 | Crystal structure of XerH site-specific recombinase bound to difH substrate: pre-cleavage complex | http://www.rcsb.org/pdb/search/structid-Search.do?structureId=5JK0 | Publicly available at the RCSB Protein Data Bank (accession no: 5JK0) |
| Bebel A, Barabas O | 2016 | Crystal structure of XerH site-specific recombinase bound to palindromic difH substrate: post-cleavage complex | http://www.rcsb.org/pdb/search/structid-Search.do?structureId=5JJV | Publicly available at the RCSB Protein Data Bank (accession no: 5JJV) |

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
