## [Decision Letter]

Thank you for submitting your article "Structural snapshots of Xer recombination reveal activation by synaptic complex remodeling and DNA bending." for consideration by *eLife*. Your article has been reviewed by two peer reviewers, and the evaluation has been overseen by a Reviewing Editor and John Kuriyan as the Senior Editor. The following individual involved in review of your submission has agreed to reveal his identity: Gregory D Van Duyne.

The reviewers have discussed the reviews with one another and the Reviewing Editor has drafted this decision to help you prepare a revised submission.

Understanding the mechanism of Xer recombination at chromosomal dif sites is acknowledged to be a long-standing, challenging problem. In this regard, the manuscript provides an extensive structural and biochemical analysis of the Helicobacter Xer-dif recombination machine in a well-written and well-documented manner, and that it reveals new details of the steps involved in this important reaction. Showing for the first time high resolution structures of a Xer recombinase complexed with DNA and these structures was unanimously considered to be useful for modelling and understanding other Xer recombinases.

This said, there was some debate as to the novelty and impact of the insights they propose. One reviewer felt that the model in which inactive, un-bent synaptic Xer/dif complexes form and are then induced to bend and cleave by FtsK is intriguing and would constitute a major advance in understanding Xer/dif recombination (if correct). Another reviewer mentioned that the structures of the pre-cleavage (and catalytically inactive) and post-cleavage (on a nicked 'suicide' substrate) synaptic complexes are important and provide some surprises, but that their mechanistic significance is more incremental, with the final model (at least as depicted) adding little to what has been published previously.

In going forward, attention should be paid to depicting and improving the discussion of what new insights can gained from the present work to resolve this debate – i.e., how specifically do the structural snapshots and associated biochemistry inform or change the field's understanding of Xer recombination mechanism in a meaningful way? In addition, there are several other issues that need to be addressed before a final decision can be reached. These include:

1) The paper overreaches in trying to reinforce the biological significance of the work. In the Abstract for example, the statement that the work will "open up new avenues towards the antibacterial treatment" is far-fetched. The same applies to the statement (in the Abstract and elsewhere) that the work provides mechanistic insight into the action of FtsK in stimulating the reaction. Similar overreach is present in the Discussion. In general, statements relating to how FtsK activates Helicobacter Xer recombination or how the present study will aid drug discovery should be downplayed, as there are no direct experiments or data that relate to these points. Relevant to the FtsK, would it be appropriate to bring up and cite the subsequent Zawadzki work (May et al. 2015) on FtsK activation as monitored in single-molecule FRET.

2) Figure 3. Was the Xer recombination observed in *E. coli* FtsK-dependent (Figure 3 and supporting information)? If the authors still wish to focus on the importance of FtsK in activating Xer recombination, this is an important experiment. If it is FtsK-dependent, is this expected given the differences between Helicobacter FtsK and *E. coli* FtsK (and would one expect Helicobacter Xer to be acted on by *E. coli* FtsK, if indeed this is the case?). Some comment on this question is needed.

3) In Figure 6 and the associated text, there is a problem with the proposal that the binding of the two recombinases to dif is temporally sequential in a mechanistic sense. How can this be? The observed affinities for the two dif arms are presumably determined by the relative off-rates, meaning that the left site is more often occupied than the right site, and leading to the second molecule binding to the right site more often than vice versa. Why not simply restate this way? Along these lines, a substantial part of Figure 6 is a repeat of what is in Figure 1. It is suggested that Figure 6 be deleted and that relevant new panels be incorporated into Figure 5.

4) The evidence that XerH does not cleave linear dif sites shown in supplemental Figure 6 seems weak. *E. coli* XerCD can cleave dif sites in vitro and in vivo in the absence of FtsK, but this cleavage is non-productive (XerC generates and resolves HJs or cleaves and re-ligates; XerD is inactive). By analogy, XerH bound to the left half-site of dif resembles XerC in its properties (i.e., the more active half). The authors have made the opposite assignment in their model of XerC/XerD/dif shown in Figure 5, which is consistent with current views that XerD catalyzes the first strand exchange when activated by FtsK. However, this assignment implicitly assigns XerD half to be the more active one, based on the data presented. The synaptic complex structure determined in this work might instead correspond to the alternative 'XerC cleavage' synaptic complex: the one that undergoes futile cycling. The reaction schematic shown in Figure 6 implies that the more reactive dif half-sites initiate strand exchange (with the help of FtsK) and the inactive half-sites resolve the HJ intermediate, based on the structural models and biochemistry presented. This is the opposite of the currently favored mechanism for *E. coli* Xer/dif recombination. Please comment on this point.

5) The cleaved complex confirms that Xer enzymes proceed through a Cre-like pathway once activated, with sharply bend DNA arms, etc. Unfortunately, the structure has been symmetrized to be Cre-like, possibly masking asymmetric features that may be present. In particular, it may be difficult to conclude much from the inter-subunit interactions present, since the asymmetries introduced by the right half-site have been lost. This point should be noted in the text.

6) The pre-cleavage complex was crystallized without divalent ions, at pH 5.5. The post-cleavage complex was crystallized at pH 7 in the presence of a high concentration of divalent ions. An alternative interpretation of the current results is that the observed structures reflect the crystallization conditions and that the synaptic complex is actually bent (to some extent) at netural pH with Mg^2+^ present. The authors should address this alternative explanation in the paper.

7) The cooperativity of XerH binding to dif is actually very low, given the similar half-site vs. full site binding constants. The wording in the manuscript is somewhat misleading in this respect – it is surprising how low the cooperativity is, given the extent of inter-subunit interactions on a single dif site. The authors should report what this interaction surface is and discuss the lack of strong cooperative binding in light of the structure.

8) It would be useful to report whether a bent, but un-nicked duplex substrate can be reconstructed (modeled) from the cleaved complex. This would indicate whether the higher energy bent form of dif is likely to resemble the cleaved configuration, or alternatively, whether the cleaved complex undergoes additional bending and subunit rotations that occur after cleavage has occurred.

9) The authors suggest that the suicide substrates used are cleaved by XerH because the nicks facilitate the required bend. Usually, the energetic cost of DNA bending is mostly related to unstacking of the bases, which nicking does not solve. Do the current structures explain how the nicks might be specifically allowing higher cleavage for this system – e.g., through relieving a strained backbone configuration? Or are they simply allowing the cleavage product to accumulate? Is the structure of an XerH catalytic mutant bound to nicked dif sites expected to be bent?

[Editors' note: further revisions were requested prior to acceptance, as described below.]

Thank you for resubmitting your work entitled "Structural snapshots of Xer recombination reveal activation by synaptic complex remodeling and DNA bending" for further consideration at *eLife*. Your revised article has been favorably evaluated by John Kuriyan (Senior editor), a Reviewing editor, and two reviewers.

The manuscript has been improved but there are some remaining issues that need to be addressed before acceptance, as outlined below. Note that these issues can all be dealt with through textual changes, and do not require any additional experiments.

1) The models in Figure 1, Figure 2 and Figure 6 cause some confusion as to how the structures might be interpreted in light of the XerH mechanism. Please consistently draft these figures with the strand going from 5' [on the left] to 3' [on the right] placed on 'top' in each of the views [i.e. viewing from the minor groove side in each case]. The authors should also acknowledge that an alternative model is consistent with their data in which FtsK is required to make the alternative synaptic complex and/or explain why such a model isn't favored.

2) Subsection “Modeling of *E. coli* XerC/D synaptic complex based on the XerH-difH structures”. In the absence of supporting data indicating that XerH will preferentially resolve HJs by bottom strand exchange, this claim should be removed. See also Discussion paragraph 7 in reference to this point.

3) Discussion section, paragraph one. Aside from the crystal structure reported here, there is other evidence that XerH/dif sites are assembled in an inactive synaptic complex in solution (e.g., Zawadzki et al. should be noted). Is there additional evidence for XerH/difH?

4) Authors' response #1 and related manuscript sections. What is the evidence that XerH does not cleave dif sites in vitro? This has been a tricky issue in the field, since it is hard to distinguish cleavage followed by rapid ligation from lack of cleavage. Unless the authors can provide data (e.g., using 5'-bridging phosphorothiolates) that demonstrate lack of cleavage, it would be better to take the position that it is not yet clear whether cleavage occurs in vitro. It is also difficult to explain activity in *E. coli* with such a model. In contrast, the behavior is readily explained if XerH bound to the left end behaves like XerC (as in an alternative model). Related to this is the question of whether cleavage is required for stabilization of the synaptic complex, independent of FtsK. This is true of λ int and several catalytic Cre mutants form much weaker synaptic complexes relative to wild-type Cre, suggesting that this could also be true in XerH as well. The nicked XerH/DNA complex also supports this idea. Thus, the synaptic complex structure observed for XerH could equally well represent a stable intermediate that can reach the transient bent configuration required for cleavage on its own. Please resolve/comment on this issue.

5) Authors' response #6. The reviewers requested that the authors note the possibility that the synaptic complex structure may be influenced by crystallization conditions (low pH and/or lack of divalents). The authors acknowledge this in their response to reviewers, but do not appear to have modified the manuscript. Given that this structure is central to the authors' proposals for Xer recombination, the readers should be made aware that the conditions for the two structures are quite different. Please revise the manuscript accordingly.

6) The authors note that several unbound Xer crystal structures show different domain relationships, but continue to promote the idea that a specific conformational change takes place upon DNA binding (as shown in the video). To echo the original review, it is unlikely that the XerD alone structure represents a true "conformational intermediate," rather than just one of an ensemble of configurations present when Xer subunits are not bound to dif. The three Xer crystal structures in the PDB have their NTDs in completely different conformations. The ideas in the discussion and video illustrating that the NTD has to undergo a significant change when completing the clamp on the dif half-site was assumed many years ago by most people in the field and is true for all tyrosine recombinases. This point should clarified or else its discussion (and the movie) should be removed from the paper.

7) The distinction between understanding the mechanism of recombination vs. regulation of this process is important, but not made clear (Abstract and Introduction sections). The authors imply that the mechanism of XerC/XerD recombination is still unknown.

8) Introduction paragraph six. The structures do not delineate the pathway (delete this) but do shed light on potential regulatory mechanisms (reword).

9) In the same section the structure does not explain why activation is required. It suggests a possible model, as described by the authors, which differs from at least one other XerCD model.

10) Results paragraph one. The current mechanism is based on much more than the series of Cre-loxP structures cited. A review covering tyrosine recombinase mechanisms (there are many) might be more appropriate here.

11) Subsection “Structure of the XerH-difH synaptic complex” need to be redrafted to accurately to describe the historical record (in doing so, the authors should consider point 9 and earlier comments regarding alternative models). The current text is incomplete and in part misleading. In the absence of FtsK, HJs formed by XerC were detected in early work and this led to the hypothesis that the XerC active conformation formed preferentially and FtsK switched the conformation of an XerC active complex to one in which XerD was active. Single-molecule FRET experiments by Zawadzki and colleagues led to the different hypothesis that this is not the case; rather, the major synaptic complex that forms in the absence of FtsK is an 'XerD potentially active conformation,' which is catalytically dead. A small proportion of XerC active complexes nevertheless form and are responsible for the XerC HJs. Nevertheless, the dominant pre-active XerD synaptic complexes are the ones activated by XerD. This is consistent with the results here, but the historical development of these ideas should be explicitly chronicled.

12) Subsection “Structure of the XerH-difH synaptic complex” paragraph two. DNA duplex, not oligonucleotide.

---

## [Author Response]

[…] In going forward, attention should be paid to depicting and improving the discussion of what new insights can gained from the present work to resolve this debate – i.e., how specifically do the structural snapshots and associated biochemistry inform or change the field's understanding of Xer recombination mechanism in a meaningful way? In addition, there are several other issues that need to be addressed before a final decision can be reached.

We hope that our revisions have added clarity with respect to the novelty of our study. We have now included a paragraph at the beginning of Discussion to summarize the novel data and insights brought about by our work. We have also modified the text throughout, particularly the final paragraph of the discussion, to better emphasize the impact of our results for understanding of Xer recombination and tyrosine recombination more generally.

The main novel aspects of our study and their impact are summarized below for your information:

We report the first high-resolution structures of an Xer recombinase in synaptic complex with its DNA substrates. Such a structure has been long sought, as its absence prevented proper understanding of this important family of site-specific recombination systems. Structural models relying on available structures of the Cre/lox system and others left several questions unanswered. Especially with regard to the molecular mechanism of Xer activation at chromosomal dif sites, it remained unclear why Xer proteins require activation in the first place, and what the molecular assemblies and rearrangements preceding activation might look like (all previous tyrosine recombinase-DNA structures reflected macromolecular assemblies highly similar to the activated state). In this manuscript, we present two structures of the XerH-difH synaptic complex captured at distinct stages of the recombination reaction, which shed light on critical steps of the reaction, and the structural changes involved in XerH activation.

Our wild-type XerH-difH synaptic complex structure reveals a cleavage-incompetent conformation with almost straight DNA that has never been observed for tyrosine recombinases before. This structure shows that conformational changes are required for recombination activation, and suggests that an external factor may be required to facilitate the transition. Based on published evidence that XerH recombination requires FtsK in H. pylori (Debowski et al., 2012) and by analogy with the XerC/D system, it seems reasonable to predict that the need for this conformational change may be key to the regulation of Xer proteins, and that FtsK may be directly involved in facilitating this transition. While this model requires further validation, our structure will help to create testable experimental hypotheses.

We also present a post-cleavage XerH-difH synaptic complex structure (trapped using nicked suicide DNA substrates). This structure resembles previously reported post-cleavage structures of other tyrosine recombinases with strongly bent DNA, confirming that the chemistry of catalysis by Xer proteins likely resembles canonical tyrosine recombination. Comparison of our pre- and post-cleavage structures also illustrates the structural rearrangements required for activation of Xer recombination. Unexpectedly, these changes include a major rearrangement of the intermolecular protein interfaces and significant DNA bending. This is consistent with previously published smFRET data (Zawadzki et al., 2013, see below), and supports the idea that pre-formed Xer recombinase synaptic complexes can be remodeled without energetically inefficient disassembly and reassembly of the complex.

Our structures also show that strong DNA bending is not necessarily concomitant with synaptic complex assembly in tyrosine recombination, at least for the XerH-difH system.

Our structural and biochemical data reveal how a small asymmetry in the *dif_H_* site helps to establish correct ordering of strand cleavages and ligations in recombination by the homomeric Xer recombinase XerH.

Modeling extends our data to the prototypical XerC/D system, indicating that the XerH-*dif_H_* structures provide a useful resource for understanding other Xer recombinases.

*1) The paper overreaches in trying to reinforce the biological significance of the work. In the Abstract for example, the statement that the work will "open up new avenues towards the antibacterial treatment" is far-fetched. The same applies to the statement (in the Abstract and elsewhere) that the work provides mechanistic insight into the action of FtsK in stimulating the reaction. Similar overreach is present in the Discussion. In general, statements relating to how FtsK activates Helicobacter Xer recombination or how the present study will aid drug discovery should be downplayed, as there are no direct experiments or data that relate to these points. Relevant to the FtsK, would it be appropriate to bring up and cite the subsequent Zawadzki work (May et al. 2015) on FtsK activation as monitored in single-molecule FRET.*

In accord with the comments of the reviewers, we have removed all the comments regarding the potential use of our structures for antibiotic discovery. We have also rephrased the text referring to FtsK activation of Xer recombination, to clarify how far our data go and where hypothesis starts.

We do however maintain that our structural and biochemical data provide some important insights into the activation of XerH recombination by FtsK. First, our observations that wild-type XerH-*dif_H_* synaptic complexes assemble in a cleavage-incompetent conformation and that XerH alone is unable to cleave wild-type DNA substrates in vitro indicates that the help of an external factor is needed for activation. In agreement with this idea, literature evidence from Debowski et al., 2012 showed that XerH recombination is critically dependent on FtsK in *H. pylori*, and strains lacking the C-terminal part of FtsK show a strong filamentous phenotype and no detectable XerH-*dif_H_* recombination. Therefore, it seems reasonable to presume that FtsK activates XerH in vivo in *H. pylori,* as *E. coli* FtsK activates XerC/D-*dif* recombination in *E. coli.* Second, comparison of the inactive XerH-*dif_H_* pre-cleavage complex with our activated post-cleavage complex structure indicates that XerH activation involves large conformational changes in pre-formed synaptic complexes. This is also in line with smFRET data reported by Zawadzki et al., 2013 and Diagne et al., 2014 for XerC/D-*dif* complexes.

Thus, we predict that FtsK is directly involved in activating XerH in *H. pylori*, probably via facilitating the conformational changes observed in our structures. Nevertheless, we acknowledge that our data do not provide clear-cut evidence for the exact function of FtsK, and do not address exactly how FtsK acts on the molecular or structural level. Insights into these questions will require further investigations, and we have now clarified this further in the manuscript (Discussion section).

Reference to May et al., 2015 has been added (Introduction section).

*2) Figure 3. Was the Xer recombination observed in E. coli FtsK-dependent (Figure 3 and supporting information)? If the authors still wish to focus on the importance of FtsK in activating Xer recombination, this is an important experiment. If it is FtsK-dependent, is this expected given the differences between Helicobacter FtsK and E. coli FtsK (and would one expect Helicobacter Xer to be acted on by E. coli FtsK, if indeed this is the case?). Some comment on this question is needed.*

The experiment shown in Figure 3 used a standard *E. coli* lab strain with wild type FtsK. To check if XerH-*dif_H_* recombination was dependent on *E. coli* FtsK in this assay, we tested recombination in the *E. coli* strain DS9041, which lacks the C-terminal part of FtsK (FtsK-C) (kindly provided by Prof. Dave Sherratt). Note that full deletion of FtsK is lethal and cannot be used in these assays, but FtsK-C deletion was previously shown to be sufficient to abolish XerC/D activation (*Recchia et al., 1999; Steiner et al., 1999*). Plasmid-based intramolecular XerH-*dif_H_* recombination assays in this strain showed similar levels of *dif_H_*-galK-*dif_H_* cassette excision as was observed in the wild-type strain, suggesting that deletion of FtsK-C does not affect this reaction.

Chromosome resolution by XerH-*dif_H_* recombination in its native *H. pylori* is known to require FtsK (*Debowski et al., 2012*). So, to check if chromosome dimer resolution by XerH-*dif_H_* recombination can occur in *E. coli*, we replaced the *dif* site of the *E. coli* chromosome with *dif_H_*-Km and supplemented this strain with an XerH expression plasmid. This showed a phenotype consistent with loss of chromosome dimer resolution (~10% of elongated *E. coli* cells that are unable to divide), irrespective of the presence or absence of *E. coli* FtsK-C. This showed that XerH-*dif_H_* recombination on chromosomes is compromised in *E. coli*, and *E. coli* FtsK cannot fully complement for the function of *H. pylori* FtsK. These results are consistent with previous reports showing that the Xer-FtsK interactions can be species-specific (*Yates et al. 2003*).

The fact that we still observe detectable levels of intramolecular XerH-*dif_H_* recombination on plasmid-borne *dif_H_*-galK-*dif_H_* cassettes in *E. coli* may be due to the high sensitivity of this assay (using a protein overexpression plasmid and multi-copy reporter plasmids), or it may indicate that factors other than *E. coli* FtsK-C (e.g. DNA bending by supercoiling, or other *E. coli* proteins) can activate XerH in this setting. However, elucidating the exact function of *H. pylori* FtsK and the effects of other interaction partners on XerH recombination requires further investigations, which are beyond the scope of the current study.

*3) In Figure 6 and the associated text, there is a problem with the proposal that the binding of the two recombinases to dif is temporally sequential in a mechanistic sense. How can this be? The observed affinities for the two dif arms are presumably determined by the relative off-rates, meaning that the left site is more often occupied than the right site, and leading to the second molecule binding to the right site more often than vice versa. Why not simply restate this way?*

We agree with the reviewer on this point. We have rephrased the corresponding text (Discussion section) and figure legend as requested, and avoid using the term ‘sequential binding’ throughout.

*Along these lines, a substantial part of Figure 6 is a repeat of what is in Figure 1. It is suggested that Figure 6 be deleted and that relevant new panels be incorporated into Figure 5.*

We agree that Figure 6 was unnecessarily repetitive. We have now removed panels E-G and keep only the parts that are needed to illustrate the model we propose for XerH recombination. For the benefit of the broad readership of *eLife*, however, we prefer to keep this model figure separate from the other data figures.

*4) The evidence that XerH does not cleave linear dif sites shown in supplemental Figure 6 seems weak. E. coli XerCD can cleave dif sites* in vitro *and* in vivo *in the absence of FtsK, but this cleavage is non-productive (XerC generates and resolves HJs or cleaves and re-ligates; XerD is inactive). By analogy, XerH bound to the left half-site of dif resembles XerC in its properties (i.e., the more active half). The authors have made the opposite assignment in their model of XerC/XerD/dif shown in Figure 5, which is consistent with current views that XerD catalyzes the first strand exchange when activated by FtsK. However, this assignment implicitly assigns XerD half to be the more active one, based on the data presented. The synaptic complex structure determined in this work might instead correspond to the alternative 'XerC cleavage' synaptic complex: the one that undergoes futile cycling. The reaction schematic shown in Figure 6 implies that the more reactive dif half-sites initiate strand exchange (with the help of FtsK) and the inactive half-sites resolve the HJ intermediate, based on the structural models and biochemistry presented. This is the opposite of the currently favored mechanism for E. coli Xer/dif recombination. Please comment on this point.*

We agree with the referee that assignment of XerC and XerD to the individual XerH subunits in our XerH-*dif_H_* synaptic complexes is ambiguous. Our XerH cleavage experiments might indeed be reporting on ‘futile cycling’ by the subunit bound at the left arm and we cannot fully exclude the possibility that our structures reflect a ‘futile cycling’ state. However, we note that currently there is no evidence that XerH is subject to the same ‘futile cycling’ mechanism as the XerC/D-*dif* system. (In fact, XerC/D itself is also not subject to this phenomenon when acting on plasmid-borne *cer* and *psi* sites.) Instead, our results indicate that while first XerH cleavage preferentially occurs on the left arm, resolution of a Holliday Junction substrate (mentioned in the Discussion section as “data not shown”) occurs by cleavage of the right *dif_H_* arm. Drawing analogy with the XerC/D-*dif* system, this observation would map XerC to the XerH subunit bound to the right arm (because it is XerC that resolves preformed HJ substrates with or without FtsK), and thus XerD to the left arm-bound subunit. This, together with the observed lack of cleavage on native (un-nicked) substrates (Figure 6—figure supplement 1), indicates that XerH might not be affected by futile cycling. Instead, XerH synaptic complexes appear to assemble directly in the correct configuration, but normally remain inactive in the absence of FtsK. Introduction of nicks in the DNA substrates might mimic the effect of FtsK, allowing activation of left arm cleavage as seen in our cleavage assays and the post-cleavage structure.

We now state this ambiguity and explain our choice for XerC/D assignment (subsection “Modeling of E. coli XerC/D synaptic complex based on the XerH-difH structures”).

*5) The cleaved complex confirms that Xer enzymes proceed through a Cre-like pathway once activated, with sharply bend DNA arms, etc. Unfortunately, the structure has been symmetrized to be Cre-like, possibly masking asymmetric features that may be present. In particular, it may be difficult to conclude much from the inter-subunit interactions present, since the asymmetries introduced by the right half-site have been lost. This point should be noted in the text.*

We appreciate that our synaptic complex with symmetrized *dif_H_*LP might not fully reflect the subunit arrangement and interactions in the native XerH-*dif_H_* complex. However, we note that the overall structure of each XerH subunit and their interactions with the respective *dif_H_* DNA arms in the XerH-*dif_H_*LP complex and the XerH – native *dif_H_* (pre-cleavage) complex are very similar (see subsection “Conformational rearrangements in the post-cleavage synaptic complex”), suggesting that substrate symmetrization does not have a profound effect on the synaptic complex structure. We note that crystal structures of Cre/lox synaptic complexes also showed highly similar arrangements with symmetrized and native substrates (Guo et al. 1997, Guo et al. 1999, Ennifar et al. 2003). Moreover, we would like to emphasize that – despite the use of symmetrized substrates – the conformations of the two XerH subunits bound to the two arms of one difHLP site are quite different and the XerH-difHLP structure is actually ‘asymmetric’ in this sense (i.e. the complex has only dyad symmetry connecting the two difH sites across the synaptic interface; there is no symmetry connecting the protein molecules bound to two halves of one difHLP site). Since we observe very similar ‘asymmetry’ in the native pre-cleavage XerH-difH complex, this feature appears to be intrinsic to the natural system. We have now revised the text (last paragraph in subsection “Conformational rearrangements in the post-cleavage synaptic complex”) to make this clearer.

*6) The pre-cleavage complex was crystallized without divalent ions, at pH 5.5. The post-cleavage complex was crystallized at pH 7 in the presence of a high concentration of divalent ions. An alternative interpretation of the current results is that the observed structures reflect the crystallization conditions and that the synaptic complex is actually bent (to some extent) at netural pH with Mg^2+^ present. The authors should address this alternative explanation in the paper.*

We acknowledge that the crystallization conditions may have an influence. However, we have tested XerH activity under the exact crystallization condition used for the pre-cleavage complex (pH 5.5, no metal ions) and observe a good level of DNA cleavage with nicked DNA substrates, indicating that the crystallization conditions are not primarily responsible for the inactive conformational state we observe in our structure. In the Cre/*lox* system, both pre- and post-cleavage synaptic complexes invariably show a similar DNA bend and inter-subunit arrangement, irrespective of the crystallization conditions which included pH of 5 – 7.6 and various divalent ion (Mg^2+^ or Ca^2+)^ concentrations.

*7) The cooperativity of XerH binding to dif is actually very low, given the similar half-site vs. full site binding constants. The wording in the manuscript is somewhat misleading in this respect – it is surprising how low the cooperativity is, given the extent of inter-subunit interactions on a single dif site. The authors should report what this interaction surface is and discuss the lack of strong cooperative binding in light of the structure.*

All binding constants mentioned in the manuscript were measured with full-length DNA substrates; the substrates *dif_H_*L and *dif_H_*R contained random DNA sequence in place of the second *dif_H_* arm. Since these substrates can still bind two XerH subunits (Figure 3—figure supplement 1) and all complex species were pooled for K_D_ calculations, the reported values do not reflect true ‘half-site’ binding constants. We apologize for this confusion, and have now revised the manuscript text to make this clear (subsection “difH asymmetry dictates the order of binding and cleavage events”).

To evaluate cooperativity, we have prepared Hill plots based on our EMSA data, by plotting the logarithm of the fractional binding of the DNA substrate, against the logarithm of the protein concentration. This gave a high value for cooperativity, in agreement with the very sharp transition of all unbound DNA to all bound (in dimeric complex) observed on our EMSA gels with native *dif_H_* (Figure 3—figure supplement 1). Similarly, high cooperativity was observed for XerC/D – *dif* binding (e.g. Cao et al., 1997 and Spiers and Sherratt, 1997), and is consistent with the extensive inter-subunit interactions observed in our structures (now reported in Discussion section paragraph four), as also pointed out by the referees. However, we are aware that the exact level of cooperativity cannot be evaluated from our data and have rephrased the corresponding sections in the text.

*8) It would be useful to report whether a bent, but un-nicked duplex substrate can be reconstructed (modeled) from the cleaved complex. This would indicate whether the higher energy bent form of dif is likely to resemble the cleaved configuration, or alternatively, whether the cleaved complex undergoes additional bending and subunit rotations that occur after cleavage has occurred.*

We have performed the modeling as recommended, and find that the broken DNA strand can be joined in the cleaved complex structure with acceptable geometry, to reconstruct a bent but un-nicked substrate. However, the modeled DNA strand clashes with the protein in a loop region just before helix αN. This is now stated in the manuscript (paragraph two subsection “Conformational rearrangements in the post-cleavage synaptic complex”). This observation suggests that the protein might block proper bending of intact DNA, and changes in its conformation may be required to enable full DNA bending. Since the affected protein segment is closely connected to the catalytic tyrosine (on helix αN), its movement – perhaps facilitated by interactions with FtsK – may even explain the direct link between DNA bending and catalytic activation. Alternatively, cleavage might occur on partially bent DNA substrates, with additional bending occurring after cleavage. Interestingly, with the nicked substrates, the clash can be avoided thanks to the missing phosphate groups (see also our response to point 9).

While these hypotheses are intriguing, further experiments are required to evaluate the actual order of DNA bending and cleavage events.

*9) The authors suggest that the suicide substrates used are cleaved by XerH because the nicks facilitate the required bend. Usually, the energetic cost of DNA bending is mostly related to unstacking of the bases, which nicking does not solve. Do the current structures explain how the nicks might be specifically allowing higher cleavage for this system – e.g., through relieving a strained backbone configuration? Or are they simply allowing the cleavage product to accumulate? Is the structure of an XerH catalytic mutant bound to nicked dif sites expected to be bent?*

Modeling of a putative bent, but un-cleaved XerH-*dif_H_* complex (based on the post-cleavage complex structure as described in point 8) indicated that the DNA might have somewhat strained backbone geometry, and the DNA appears to clash with the protein at the phosphate group exactly one nucleotide downstream of the scissile phosphate. Thus, we believe that the nicks favor the bent DNA conformation (paragraph two subsection “Conformational rearrangements in the post-cleavage synaptic complex”) by helping to increase the flexibility of the DNA strands at the bending points and relieving the strained backbone conformation, and they may also help avoid unfavourable interactions with the protein upon DNA bending due to the absence of the phosphate implicated in the clash. Accordingly, we would expect a catalytic mutant XerH to bind to nicked *dif_H_* substrate roughly in the same conformation as the wild type protein, with a sharp DNA bend.

[Editors' note: further revisions were requested prior to acceptance, as described below.]

*[…] 1) The models in Figure 1, Figure 2 and Figure 6 cause some confusion as to how the structures might be interpreted in light of the XerH mechanism. Please consistently draft these figures with the strand going from 5' [on the left] to 3' [on the right] placed on 'top' in each of the views [i.e. viewing from the minor groove side in each case]. The authors should also acknowledge that an alternative model is consistent with their data in which FtsK is required to make the alternative synaptic complex and/or explain why such a model isn't favored.*

We appreciate this comment and understand the reviewers’ concern regarding our drawings. Unfortunately, presenting the structures with viewing from the minor groove side would obscure visibility of important features of the structures (such as the DNA and the swapped C-terminal helix). Moreover, to conform with the previously published nomenclature of the *difH* arms we prefer to keep the ‘left arm’ on the left and the ‘right arm’ on the right side in our figures. This means that the schematic models in Figure 1 and Figure 6 should be drawn with the strand going 3’ to 5’ on the top to be consistent with the structure figures. However, to address the reviewers’ concern and avoid confusion, we have now inverted the representation of the DNA sequence in Figure 2 (as well as in Figure 4 and in the figure supplements) so that it is now consistent with all other figures. In addition, we have clearly labeled both 5’ and 3’ ends of both strands on all DNA molecules in all drawings (see Figure 1, Figure 2 and Figure 6), and state the directionality of the DNA in the corresponding legends.

Concerning the possible alternative model for the role of FtsK, this possibility is now mentioned and discussed in the manuscript (Discussion section paragraph five).

*2) Subsection “Modeling of E. coli XerC/D synaptic complex based on the XerH-difH structures”. In the absence of supporting data indicating that XerH will preferentially resolve HJs by bottom strand exchange, this claim should be removed. See also Discussion paragraph 7 in reference to this point.*

All claims referring to HJ resolution by XerH have now been removed from the manuscript as requested.

*3) Discussion section, paragraph one. Aside from the crystal structure reported here, there is other evidence that XerH/dif sites are assembled in an inactive synaptic complex in solution (e.g., Zawadzki et al. should be noted). Is there additional evidence for XerH/difH?*

This point has now been noted in the text. For the XerH/difH system in particular, apart from our structural and biochemical data, we are not aware of any additional literature evidence showing that synaptic complexes initially assemble in an inactive conformation.

*4) Authors' response #1 and related manuscript sections. What is the evidence that XerH does not cleave dif sites* in vitro*? This has been a tricky issue in the field, since it is hard to distinguish cleavage followed by rapid ligation from lack of cleavage. Unless the authors can provide data (e.g., using 5'-bridging phosphorothiolates) that demonstrate lack of cleavage, it would be better to take the position that it is not yet clear whether cleavage occurs* in vitro*. It is also difficult to explain activity in E. coli with such a model. In contrast, the behavior is readily explained if XerH bound to the left end behaves like XerC (as in an alternative model). Related to this is the question of whether cleavage is required for stabilization of the synaptic complex, independent of FtsK. This is true of λ int and several catalytic Cre mutants form much weaker synaptic complexes relative to wild-type Cre, suggesting that this could also be true in XerH as well. The nicked XerH/DNA complex also supports this idea. Thus, the synaptic complex structure observed for XerH could equally well represent a stable intermediate that can reach the transient bent configuration required for cleavage on its own. Please resolve/comment on this issue.*

The reviewer is concerned about the possibility of “futile” DNA cleavage and ligation in the XerH-*difH* system. We have no evidence that this occurs in the XerH-*difH* system, but we cannot exclude the possibility based on our data. We now discuss the alternative possibilities raised by the reviewers in the manuscript.

*5) Authors' response #6. The reviewers requested that the authors note the possibility that the synaptic complex structure may be influenced by crystallization conditions (low pH and/or lack of divalents). The authors acknowledge this in their response to reviewers, but do not appear to have modified the manuscript. Given that this structure is central to the authors' proposals for Xer recombination, the readers should be made aware that the conditions for the two structures are quite different. Please revise the manuscript accordingly.*

We now point out the difference in crystallization conditions in the manuscript as requested.

*6) The authors note that several unbound Xer crystal structures show different domain relationships, but continue to promote the idea that a specific conformational change takes place upon DNA binding (as shown in the video). To echo the original review, it is unlikely that the XerD alone structure represents a true "conformational intermediate," rather than just one of an ensemble of configurations present when Xer subunits are not bound to dif. The three Xer crystal structures in the PDB have their NTDs in completely different conformations. The ideas in the discussion and video illustrating that the NTD has to undergo a significant change when completing the clamp on the dif half-site was assumed many years ago by most people in the field and is true for all tyrosine recombinases. This point should clarified or else its discussion (and the movie) should be removed from the paper.*

The respective discussion and movie have been removed in the current version of the manuscript.

*7) The distinction between understanding the mechanism of recombination vs. regulation of this process is important, but not made clear (Abstract and Introduction section). The authors imply that the mechanism of XerC/XerD recombination is still unknown.*

The last sentence of the Abstract has been rephrased to better distinguish between catalytic mechanism and its regulation; and the sentence that was in the Introduction has been deleted to avoid incorrect implications.

*8) Introduction paragraph six. The structures do not delineate the pathway (delete this) but do shed light on potential regulatory mechanisms (reword).*

Corrected.

*9) In the same section the structure does not explain why activation is required. It suggests a possible model, as described by the authors, which differs from at least one other XerCD model.*

Rephrased.

*10) Results paragraph one. The current mechanism is based on much more than the series of Cre-loxP structures cited. A review covering tyrosine recombinase mechanisms (there are many) might be more appropriate here.*

Corrected.

*11) Subsection “Structure of the XerH-difH synaptic complex” need to be redrafted to accurately to describe the historical record (in doing so, the authors should consider point 9 and earlier comments regarding alternative models). The current text is incomplete and in part misleading. In the absence of FtsK, HJs formed by XerC were detected in early work and this led to the hypothesis that the XerC active conformation formed preferentially and FtsK switched the conformation of an XerC active complex to one in which XerD was active. Single-molecule FRET experiments by Zawadzki and colleagues led to the different hypothesis that this is not the case; rather, the major synaptic complex that forms in the absence of FtsK is an 'XerD potentially active conformation,' which is catalytically dead. A small proportion of XerC active complexes nevertheless form and are responsible for the XerC HJs. Nevertheless, the dominant pre-active XerD synaptic complexes are the ones activated by XerD. This is consistent with the results here, but the historical development of these ideas should be explicitly chronicled.*

We now describe the historical development of the available models for XerC/D activation in the current version of the manuscript as requested. (Results section first paragraph)

*12) Subsection “Structure of the XerH-difH synaptic complex” paragraph two. DNA duplex, not oligonucleotide.*

Done.